# Harmonised boundary layer wind profile dataset from six ground-based doppler wind lidars in a transect across Paris, France

William Morrison[1], Dana Looschelders[1], Jonnathan Céspedes[2], Bernie Claxton[3], Marc-Antoine Drouin[4], Jean-Charles Dupont[5], Aurélien Faucheux[6], Martial Haeffelin[7], Christopher C. Holst[8], Simone Kotthaus[2], Valéry Masson[9], James McGregor[3], Jeremy Price[3], Matthias Zeeman[1], Sue Grimmond[10,] Andreas Christen[1]

[1]Chair of Environmental Meteorology, Institute of Earth and Environmental Sciences, Faculty of Environment and Natural Resources, University of Freiburg, Freiburg, 79085, Germany
[2]Laboratoire de Météorologie Dynamique (LMD-IPSL), École Polytechnique, Institut Polytechnique de Paris, Palaiseau CEDEX, France
[3]Met Office, Exeter, EX1 3PB, UK
[4]LMD/IPSL, École Polytechnique, Institut Polytechnique de Paris, ENS, Université PSL, Sorbonne Université, CNRS, Palaiseau France
[5]Institut Pierre Simon Laplace (IPSL), Université Versailles Saint-Quentin-en Yvelines, Palaiseau Cedex, France
[6]CEREA, ENPC, EDF R&D, Institut Polytechnique de Paris, Champs-sur-Marne, France
[7]Institut Pierre Simon Laplace (IPSL), CNRS, Ecole polytechnique, Institut Polytechnique de Paris, 91128 Palaiseau CEDEX, France
[8]Institute of Meteorology and Climate Research – Atmospheric Environmental Research (IMK-IFU), Karlsruhe Institute of Technology (KIT), 82467, Germany
[9]Centre National de Recherches Météorologiques, University of Toulouse, Toulouse, 31057, France
[10]Department of Meteorology, University of Reading, Reading, RG6 6ET, United Kingdom

*Correspondence to*: Andreas Christen (andreas.christen@meteo.uni-freiburg.de)

**Abstract.** Doppler wind lidars (DWL) offer high-resolution wind profile measurements that are valuable for understanding atmospheric boundary layer (ABL) dynamics. Here six ground-based DWL, deployed in a multi-institutional effort along a 40 km transect through the centre of Paris (France), are used to retrieve horizontal wind speed and direction through the ABL at 18 - 25 m vertical and 1- 60 min temporal resolution. Data are available for June 2022 – March 2024 (three DWL) and two Intensive Observation Periods (six DWL) across 9 weeks in September 2023 – December 2023. Data from all sensors are harmonised in terms of quality control, file format, as well as temporal and vertical resolutions. The quality of this DWL dataset is evaluated against *in-situ* measurements at the Eiffel Tower and radiosonde profiles. This unique, spatially dense, open dataset will allow urban boundary layer dynamics to be explored in process-studies, and is further valuable for the evaluation of high-resolution weather, climate, inverse and air pollution models that resolve city-scale processes.

## 1. Introduction

There is a growing need for atmospheric observation networks that capture urban weather and climate phenomena at high spatial and temporal resolutions (Grimmond et al., 2010; Baklanov et al., 2018). With some numerical weather prediction (NWP) models now having horizontal grid-resolutions of $O$1 km globally (Wedi et al. 2020) and of the $O$0.1 km regionally (Lean et al., 2019), cities are increasingly well captured by these simulations. In turn, this requires a greater density of observations in order to understand the spatial variability across a city that could be expected (e.g. Fenner et al. 2024). Further, as cities look towards sustainable, net-zero futures, high spatial and temporal resolution

wind observations are crucial when considering the dispersion of urban pollutants including for inverse modelling of

greenhouse gas emissions at the city scale (e.g. Staufer et al., 2016; Che et al., 2022; Lian et al., 2023), building

construction and wind gust risk (Kent et al., 2017), wind energy yields (Stathopoulos et al., 2018) and urban-scale

heat exposure (e.g. Lemonsu et al., 2024).

Observations of wind are challenging to conduct in cities due to the nature of the roughness elements. A standard

World Meteorological Organization (WMO) in-situ wind measurement at 10 m above ground level (Liu et al., 2023;

WMO, 2024) typically is located within the roughness sublayer and hence directly influenced by the surrounding

roughness elements (Lane et al., 2013). With ground-based Doppler wind lidars (DWL) commercially available, high

resolution wind profiles through the atmospheric boundary layer (ABL) are possible (Kotthaus et al., 2023).

DWL wind profiles have been used to evaluate urban roughness parameterisations (e.g. Kent et al., 2017), wind gust

parametrisations (e.g. Kent et al., 2018), urban NWP (e.g. Fenner et al., 2024; Lean et al., 2019; Pentikäinen et al.,

2023) and large eddy simulation (LES) of urban wind fields under neutral atmospheric conditions (e.g. Filioglou et

al., 2022). These data have resolved fundamental ABL processes such as low-level jets in urban areas (Barlow et al.,

2015; Céspedes et al., 2024; Fenner et al., 2024; Zeeman et al., 2022) and tall building wakes (Theeuwes et al., 2024)

that are challenging to measure. As model complexity and resolution increase, well-documented observations are

needed from multiple locations across the urban-rural continuum and under different synoptic conditions (i.e. long,

seasonally varying time series for various land-use types and different urban densities), in standardised, accessible

data formats.

In this paper we present a harmonised dataset of simultaneously observed horizontal wind speed and direction from a

transect of six DWL through Paris operating between 2022 – 2024. The harmonisation process involves application

of a wind retrieval algorithm to raw instrument data files, aggregation of data to a common resolution (time and height

dimensions), and application of a unified quality control procedure.

Beyond regional applications, the six-DWL transect can help elucidate potential urban effects across Paris by

capturing urban-rural interactions and intra-urban variability. Paris is inland with relatively small orographic

variability, surrounded by a fairly homogeneous rural area. A number of projects are set to benefit from such

observations, including the ICOS-cities project aiming at measuring city-scale emissions (Christen et al., 2023), the

CATRINE activities improving inverse modelling of city-scale emissions (Che et al., 2024), the PAris region urbaN

Atmospheric observations and models for Multidisciplinary rEsearch (PANAME) initiative framework (Haeffelin et

al., 2023), the ACROSS air pollution campaign (Cantrell and Michoud, 2022), the Paris 2024 Olympics Research

Development Project (RDP) (https://www.umr-cnrm.fr/RDP_Paris2024), the CORDEX URBan environments and

Regional Climate Change (URB-RCC, Langendijk et al., 2024) and the *urbisphere* project (Fenner et al., 2024;

Morrison et al., 2023).

## 2. Doppler wind lidar measurement principles

### 2.1. Theoretical background

Ground-based DWL have a laser that emits light at a specified wavelength into the atmosphere. This light propagates through the atmosphere and scatters after interaction with atmospheric aerosols and cloud droplets. The motion of aerosols along the beam imparts a Doppler shift on the scattered light, causing the return signal to be shifted in frequency relative to the emitted pulse (Liu et al., 2019). The magnitude of this frequency shift directly relates to the motion of the particles that scattered the light back, which in turn is associated with the radial velocity: the component of the wind along the line of sight at a given distance (range) from the DWL. Thousands of pulses (pulse integration count) are needed to be able to determine a statistically weighted velocity. The maximum range is typically up to 12 km but can vary by instrument manufacturer, model or serial number.

### 2.2. Scan configurations

DWLs retrieve horizontal and vertical wind components in the ABL through various carefully designed scanning configurations with the following parameters: azimuth ($\theta$) and zenith ($\varphi$) emission angles of the laser, number of unique ($\theta$, $\varphi$) angles within one complete scan, range resolution at which the atmosphere is probed along the laser beam (range gate resolution, m) which – along with any oversampling – determines the maximum vertical resolution (Held and Mann, 2018), and temporal resolution. There are two scan configurations used in this dataset:

● *Velocity Azimuth Display* (VAD) uses beams at one fixed zenith angle that rotates around typically 6 - 24 azimuth angles. The measured radial velocities across all azimuth angles for a given range gate are used to retrieve the three wind components by e.g. sine wave fitting (Browning and Wexler, 1968; Weitkamp, 2005) or by least-squares fitting in matrix form (Päschke et al., 2015; Teschke and Lehmann, 2017). The average horizontal wind direction and speed for the conical scan geometry are then calculated from the wind components.

● *Doppler Beam Swinging* (DBS) (Röttger et al., 1978), a simplified VAD with fewer azimuth angles, allows faster wind profile sampling rates (Rahlves et al., 2022; Wildmann et al., 2020). The fewer azimuth samples (typically 4 cardinal and one vertical direction are sampled in one full DBS scan) allows for higher temporal resolution retrievals in an effort to capture unsteady flows (e.g. in urban areas) more completely (Lane et al., 2013).

## 3. Methods

### 3.1. Measurement stations

Six DWLs were located along a 40 km linear transect from SW to NE (aligned 250° to 35°, from N) in the Paris region (Table 2, Figure 1), passing through the City of Paris. Each measurement station is identified by a six-letter code, with the first two letters ("PA") indicating Paris for all. Instruments were located on either high-rise (PACHEM, PAJUSS, PALUPD) or low-rise (PAROIS, PASIRT) rooftops, or at ground level (PAARBO). These stations are part of a multi-institutional network undertaking boundary layer profiling, as well as radiative and sensible heat flux measurements during the campaign period of 2022 – 2024 for multiple projects with the campaign centre of operations at the Site Instrumental de Recherche par Télédétection Atmosphérique (SIRTA) long-term observatory (Haeffelin et al., 2005).

### 3.2. Network design

The most south-westerly measurement station (PASIRT, Figure 1), is located at the SIRTA observatory (20 km from Paris). The land cover fraction within a 5 km radius (Table 1) is  predominantly institutional developments (class: discontinuous urban, 41 % cover), agriculture (26 %) and forest (11 %) on a plateau about 160 m asl (above sea level) (Haeffelin et al., 2005). The transect passes through the Paris region's suburbs (PAARBO) with a majority discontinuous urban land cover (64 %) and a surrounding park (21 % forest). The centre (PAJUSS, PALUPD) and central-NE (PACHEM) of Paris have predominantly (dis)continuous urban fabric with Aéroport Roissy-Charles-de-Gaulle (PAROIS) 23 km NE of Paris uniquely sited at an airport surrounded mainly by agricultural fields (66 % airport, road and rail; 26 % agricultural). Stations are expected to be upwind, within and downwind of the Paris built-up area (Figure 1). The transect layout is aligned with the predominate south-westerly wind directions and the less common north-easterly (Figure 1) flow, where most low-level jets have been observed (Céspedes et al., 2024).

The Paris topography (Figure 1, lines) is defined by the River Seine basin at 20 m a.s.l in the city centre, and the surrounding plateaus at up to 217 m asl (within Figure 1 extent). The City of Paris (Figure 1, dense urban) topography has 20 m – 130 m asl variation and PASIRT is on the ~160 m asl Paris-Saclay Plateau (Céspedes et al., 2024).

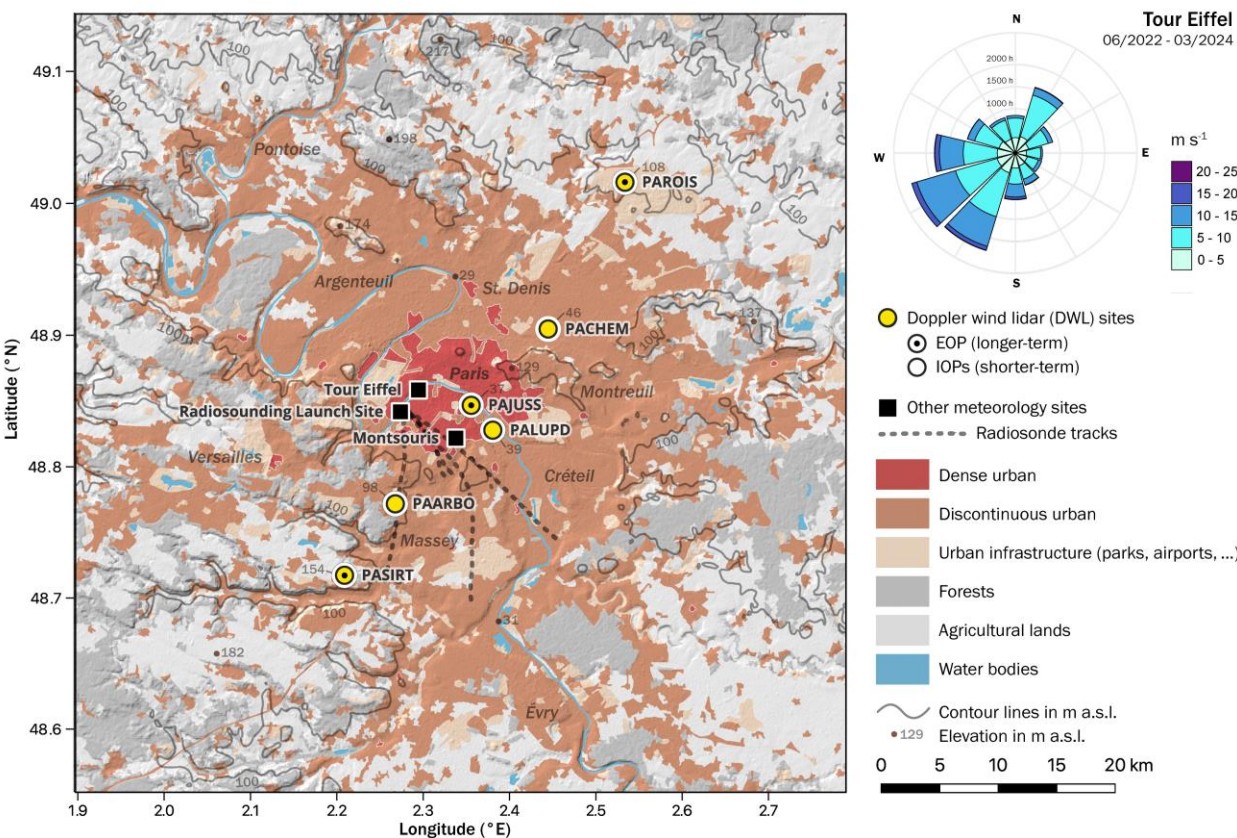

**Figure 1. Paris region land cover and orography, with location of the Doppler wind lidar (DWL) stations and other surface stations referred in this paper. In-situ wind rose  (upper right) measured at the Tour Eiffel Météo-France meteorology station at 321 m agl.**

**Table 1. Land-cover fractions in a 5-km radius around the six DWL sites for simplified classes based on the Copernicus CORINE Land Cover classification. Original CORINE land cover classes considered in each line item are given in brackets (European Environment Agency, 2020).**

| Station | PAROIS | PACHEM | PALUPD | PAJUSS | PAARBO | PASIRT |
|---|---|---|---|---|---|---|
| Continuous urban fabric (111) | 0% | 0% | 42% | 78% | 0% | 0% |
| Discontinuous urban fabric (112) | 3% | 59% | 29% | 2% | 64% | 41% |
| Industrial, commercial, construction sites (121, 133) | 5% | 23% | 11% | 6% | 5% | 12% |
| Airports, road and rail networks (122, 124) | 66% | 10% | 7% | 6% | 0% | 0% |
| Green urban areas, sport and leisure facilities (141) | 0% | 7% | 7% | 5% | 9% | 9% |
| Agricultural lands (211, 231, 242) | 26% | 1% | 0% | 0% | 0% | 26% |
| Forests (311) | 1% | 0% | 0% | 0% | 21% | 11% |
| Water bodies (511) | 0% | 0% | 3% | 3% | 0% | 0% |

### 3.3. Operation periods

The dataset, covering July 2022 to March 2024, consists of three main periods:

1. **Extensive observation period (EOP) 14/06/2022 – 31/03/2024.** The EOP objective is to capture a wide range of synoptic and seasonal weather conditions with the trade-off being a reduced, coarser spatial network of three DWLs with concurrent observations at the city centre (PAJUSS) and transect ends (PASIRT and PAROIS). PASIRT is the long-term reference station operating since 06/2009 (Haeffelin et al., 2005) (Figure 2). The PAROIS DWL long-term deployment was decommissioned on 11th December 2023 (Table 4).

2. During two **Intensive observation periods (IOP)** all six DWL have concurrent data available. **IOP1 08/08/2023 – 13/09/2023** has a range of late summer conditions, including an air pollution episode from 05/09/2023 to 08/09/2023 under south-easterly anticyclonic conditions. **IOP2 (13/11/2023 – 11/12/2023)** covers late autumn to early winter conditions, with predominantly westerly cyclonic flow. The denser network allows comparison to the EOP instruments, and observation of intra-urban variability. The three additional stations are deployed in the city centre (PALUPD) and between the city centre and transect edges (PAARBO, PACHEM). Between the two IOPs, the PAARBO sensor was down (13/09/2023 - 13/11/2023, Figure 2a). IOP2 ends when PAROIS is decommissioned, although five systems continued operation until Feb 2024.

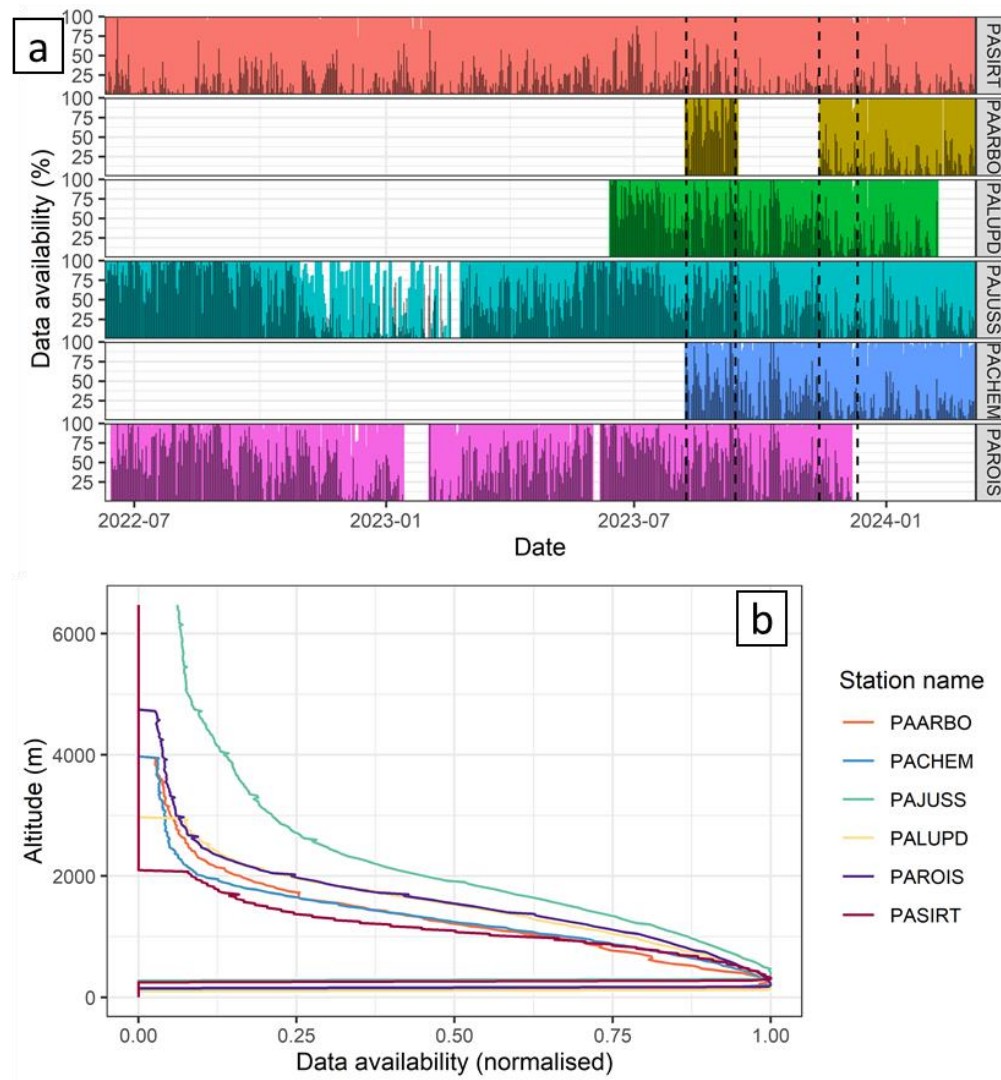

**Figure 2:** Data availability for (a) the whole extensive observation period (EOP) and intensive observation periods (IOP1 and IOP2) by station (ordered from north-east to south-west) with harmonised daily data availability as a % of maximum possible data available at 300 m agl (colour) and 1300 m agl (grey), and (b) by height (altitude) with normalised availability relative to the gate with maximum availability. The near-horizontal lines at lower altitudes indicate low/no data for the first range gates.

**Table 2:** Station locations with Doppler wind lidar sensor height (instruments details, Table 4), terrain altitude (height above sea level) based on WGS84 EGM96 Geoid determined using Google Earth Pro v7.3.6.9796 and 3D building heights (above ground level). Site owners include : Laboratoire Atmosphères et Observations Spatiales (LATMOS), Laboratoire Interuniversitaire des Systèmes Atmosphériques (LISA) and Site Instrumental de Recherche par Télédetection Atmosphérique (SIRTA) an Institut Pierre-Simon Laplace (IPSL) observatory dedicated to cloud and aerosol research. Regional location given relative to PAJUSS as the city centre (CC) reference location.

| Station code | Full station name | Lat (°N), Lon (°E) | Terrain altitude (m asl) | Instrument altitude (m asl) | Instrument height (m agl) | Siting detail: Mounting Level Building Type Site Owner/operator Site Name/ID | Operation period (DD/MM/ 20YY) | Regional location |
|---|---|---|---|---|---|---|---|---|
| PAROIS | Aéroport Roissy-Charles-de-Gaulle | 49.0160, 2.53366 | 108 | 112 | 4 | Roof: 2 storey Météo-France ROISSY site WMO ID 07157 | 14/06/22 – 11/12/23 | Airport 23 km North East of CC |

| | | | | | | | | |
|---|---|---|---|---|---|---|---|---|
| PACHEM | Chemin Vert Bobigny | 48.9046, 2.44470 | 46 | 98 | 52 | Roof: 19 storey residential building | 06/08/23 – 04/03/24 | Suburbs 10 km North East of CC |
| PAJUSS | Tour Zamansky, Jussieu | 48.8469, 2.3555 | 37 | 125 | 88 | Roof: 26 storey institutional building LATMOS, Sorbonne University QUALAIR supersite | 14/-06/2022 – 30/11/2024 | Inner CC |
| PALUPD | LISA Université Paris Diderot | 48.8278, 2.38064 | 39 | 65 | 26 | Roof: 8 storey LISA University building (Foret et al., 2022) | 29/11/22 – 07/02/24 | |
| PAARBO | Arboretum de la Vallée-aux-Loups | 48.7717, 2.26769 | 98 | 99 | 1 | Ground Arboretum maintenance yard | 27/07/23 – 05/03/24 | Suburbs 10 km South West of CC |
| PASIRT | SIRTA, IPSL, École Polytechnique | 48.7173, 2.20887 | 154 | 154 | 4.5 | Roof single storey: SIRTA, Laboratoire de Météorologie Dynamique (Dupont et al., 2016; Haeffelin et al., 2005) | 01/2011 – present | Suburbs/rural 18 km South West of CC |

### 3.4. Instrument models and measurement locations

The harmonised dataset includes observations from four different DWL instrument models (Table 3). No cross-

calibration between co-located instruments was conducted due to logistical challenges in co-locating long-term EOP

instruments with IOP instruments, instrument maintenance delays, and the prioritisation of maximising IOP data

availability.

As each instrument has a wide range of adjustable settings, this information is part of the instrument "deployment"

data (Table 4), which includes details such as physical positioning within a station, software version, and scanning

strategy.

**Table 3:** Doppler wind lidar models from different manufacturers used to collect the observational datasets. Note Halo Photonics
was acquired by the Lumibird group (Lannion, France) at the end of December 2019. Refer to Table 4 for specific instrument
deployment details. *The maximum programmable range and not necessarily the maximum range for valid radial velocity
retrievals. **StreamLine radial wind accuracy derived from **Newsom and Krishnamurthy (2022)**.

| Manufacturer | Model | Serial number | Detection bandwidth ($\pm$, m s$^{-1}$) | Doppler velocity resolution (m s$^{-1}$) | Radial wind accuracy (m s$^{-1}$) | Wavelength (μm) | Maximum range* (m) |
|---|---|---|---|---|---|---|---|
| Halo Photonics | StreamLine | 204 (METEK 0214088635) | 38 | 0.07644 | 0.1 ** | 1.55 | 12006 |
| Halo Photonics | StreamLine | 175 (METEK 0213098255) | 38 | 0.07644 | 0.1 | 1.55 | 12006 |
| Halo Photonics | StreamLine | 26 | 19.4 | 0.0191 | 0.1 | 1.55 | 3006 |
| Halo Photonics | StreamLine XR | 156 | 19.4 | 0.0382 | 0.1 | 1.55 | 12006 |
| Halo Photonics | StreamLine | 30 | 19.4 | 0.0382 | 0.1 | 1.55 | 4800 |
| Vaisala | WindCube WLS70 | 10 | 30 | 0.2 | 0.3 | 1.543 | 2000 |
| Vaisala | WindCube Scan 400S | WCS000243 | 30 | unavailable | 0.1 | 1.54 | 6750 |

### 3.4.1. Halo Photonics StreamLine instruments and deployments

Five Halo Photonics (now Lumibird group, Lannion, France) StreamLine DWLs are used (Table 3). The StreamLines report a signal-to-noise ratio SNR = S / N, with S the average signal power and N the average noise power, with SNR = 0 no signal (Päschke et al., 2015). StreamLine XR (at PAARBO) has better SNR and an extended range, compared to the non-XR StreamLine (Le et al., 2024) (stations PALUPD, PAARBO, PACHEM, PAROIS). The StreamLine's rotating scanner head allows full hemispherical coverage. These sensors have previously been deployed in urban areas (e.g. Fenner et al., 2024; Lane et al., 2013; Theeuwes et al., 2024; Yim, 2020a; Zeeman et al., 2022) often in multi-instrument campaigns. METEK GmbH, Elmshorn, Germany configured the hardware for two instruments (Table 3, serial number).

VAD scans configured on each instrument computer use scan schedule v14a.vi software and daily schedule (.dss) files. Each VAD scan has 12 equally spaced azimuth points ($\Delta\theta = 30°$) at $\varphi=15°$ with 1.4 min ± 0.1 min duration, repeating every 10 min at rounded intervals (e.g. 12:00, 12:10, 12:20, …) except for serial number (SN) 30 (at PAROIS) prior to 12[th] July 2022 that had hourly 6-point VAD. Between VAD scans the instruments stare vertically for a duration of 8.6 ± 0.1 min. DWL SN 204 (at PALUPD) had a scan schedule configuration error between Nov 2022 – Jun 2023, which led to the VAD data being corrupted and unusable for the derivation of wind direction and wind speed.

The instrument pitch and roll were levelled to 0° (±0.1°) using the internal inclinometers and the instrument bearing determined using a known hard target. As PAARBO had no hard targets available, the instrument was aligned parallel to a courtyard wall with true north determined using Google Earth Pro version 7.3.6.9796 imagery.

### 3.4.2. WindCube Scan 400S (w400S) instrument and deployment

A Vaisala Oyj (Vantaa, Finland) WindCube Scan 400S SN WCS000243 (hereafter "w400S") was deployed at PAJUSS. Some subsets of the w400S data included here are analysed by Céspedes et al. (2024). Similar Windcube Scan models have been used in other urban settings (e.g. Windcube 100S, He et al. 2021).

The w400S has lower spatial resolution than the StreamLine sensor with a first range at 150 m (here ~45 m for StreamLine, Table 5).

During this w400S deployment, from 1[st] June 2022 to 31[st] May 2024 (Table 4) at PAJUSS (Table 2), the laser pulse configuration had a spatial resolution of 75 m but the resolution of the final product is increased to 25 m through oversampling in the manufacturer retrieval algorithm. A blind zone with no wind retrievals spans over the first two measurement gates (150 m). The w400S has a rotating scanner head. Horizontal wind is retrieved by the instrument manufacturer's firmware using a five-point DBS scan (one vertical point $\varphi=0°$ and one per cardinal direction at $\varphi=15°$) taking ~ 15 s based on a 1 s accumulation time per line of sight and 2 s between scan points. The w400S is aligned to true north using a hard target with a ±2° accuracy (Céspedes et al., 2024).

### 3.4.3. Vaisala WindCube WLS70 instrument and deployment

A Vaisala Oyj WindCube WLS70 SN 10 (hereafter "WLS70") was deployed at PASIRT (Table 2) throughout the EOP. The WLS70 has a fixed 4-point DBS scan and 50 m spatial resolution (Cariou et al., 2009). Data included here are for 14th June 2022 – 31$^{st}$ March 2024 (Figure 2). As the instrument was neither moved nor modified, there is one deployment (Table 4). Subsets of the data have been formally analysed (e.g. Dupont et al., 2016; Foret et al., 2022) as the instrument is part of the long-term SIRTA observatory (Haeffelin et al., 2005).

**Table 4:** Overview of Doppler wind lidar data availability by sensor deployments for each measurement configuration, with range gate (RG) information. The w400S range gates (*) have 75 m resolution with on-board oversampling to give display resolution of 25 m. Instrument bearing corrections (clockwise from true north) are applied to both raw and final harmonised data. Number (#) of rays and range gates used are indicated.

| Station code | SN | Start date DD/MM/YY | End date DD/MM/YY | # RG | RG length (m) | Bearing (°) correction (raw/final) | VAD pulse integration count | Horizontal wind sample rate (s) | Horizontal wind scan type and zenith (φ) angle: (# of rays per scan) | Focus (m) | Comments |
|---|---|---|---|---|---|---|---|---|---|---|---|
| PAARBO | 156 | 07/08/23 | 13/09/23 | 223 | 18 | 9 / 9 | 50000 | 600 | VAD 15° (12) | Inf | |
| PAARBO | 204 | 13/11/23 | 05/03/24 | 223 | 18 | 9 / 9 | 50000 | 600 | VAD 15° (12) | 2000 | |
| PACHEM | 175 | 06/08/23 | 04/03/24 | 223 | 18 | 88 / 92.5 | 50000 | 600 | VAD 15° (12) | Inf | |
| PAJUSS | 243 | 04/10/22 | 29/10/22 | 265 | 25* | 0 / 0 | | 2 s for 720 s, → 480 s other scans | DBS 15° (5) | | |
| PAJUSS | 243 | 29/10/22 | 24/02/23 | 265 | 25* | 0 / 0 | | 2 s for 720 s, followed by 480 s of other scans | DBS 15° (5) | | Deployment: low laser power, typically only wind retrievals at cloud base |
| PAJUSS | 243 | 16/03/23 | 31/03/24 | 265 | 25* | 0 / 0 | | 2 s for 720 s interval, followed by 480 s of other scans | DBS 15° (5) | | Laser replaced; software updated |
| PAJUSS | 243 | 14/06/22 | 04/10/22 | 265 | 25* | 0 / 0 | | 2 s for 720 s interval, followed by up to 480 s of other scans | DBS 15° (5) | | Instrument bearing precision ±2° (Céspedes et al. 2024). Instrument not moved in subsequent deployments |
| PALUPD | 26 | 13/06/23 | 07/02/24 | 167 | 18 | 330 / 328 | 50000 | 600 | VAD 15° (12) | 500 | |
| PAROIS | 30 | 14/06/22 | 22/06/22 | 200 | 24 | 212 / 207 | 15000 | 1800 | VAD 15° (6) | Inf | Instrument default VAD scan |
| PAROIS | 30 | 22/06/22 | 08/08/22 | 200 | 24 | 212 / 207 | 15000 | 3600 | VAD 15° (6) | Inf | 6-point VAD scan sample rate reduced allowing other scans |
| PAROIS | 30 | 08/08/22 | 07/12/22 | 200 | 24 | 212 / 207 | 30000 | 3600 | VAD 15° (6) | Inf | Pulses integration count increased to improve signal-to-noise ratio |
| PAROIS | 30 | 07/12/22 | 11/12/23 | 200 | 24 | 212 / 207 | 30000 | 600 | VAD 15° (12) | Inf | Sample rate not consistent on each hour (HH:10, HH:20, …) |
| PASIRT | 10 | 14/06/22 | 31/03/24 | 40 | 50 | 0 / 0 | | 2 | DBS 15° (4) | | |

### 3.5. Data processing levels and quality control (QC) flags

The harmonisation process distinguishes between four levels of data. *Raw* data are those saved to the measurement device directly after each instantaneous or internally aggregated measurement with no post-processing or QC steps. *Level 1 (L1)* data have horizontal wind retrievals calculated from *raw*. *L1* may include manufacturer- and instrument-specific QC steps and thresholds. *L2* has non-instrument specific QC and associated flags. *L3* is the harmonised, published data product and is the aggregation of L2 to a common resolution (time and height dimensions).

Quality control (QC) and data availability are documented in the harmonised dataset using four Boolean QC flags:

1) *flag_low_signal_warn:* signal low enough for retrieval to be suspect. Values not rejected but retrieval should be used with caution.
2) *flag_low_signal_removed:* signal too low and retrieval is rejected.
3) *flag_suspect_retrieval_warn:* retrieval result is suspect and flagged (unrelated to *flag_low_signal*). Retrieval result should be used with caution.
4) *flag_suspect_retrieval_removed:* retrieval result is erroneous and flagged (unrelated to *flag_low_signal*). Retrieval result is rejected.

The flag value is 1 when the respective condition is satisfied.

Presented in the following subsections are instrument model-specific thresholds and processing steps for calculation of vertical profiles of horizontal wind QC flags.

### 3.6. Pre-harmonisation steps: data collection, wind retrieval processing, quality control (QC)

The raw data samples collected by the DWL instruments are automatically uploaded to secure remote data archives (Zeeman et al., 2024). Detail of routine instrument maintenance (e.g. cleaning) and in response to issues (e.g. instrument failure), are provided in the dataset supplement (Morrison et al., 2024).

#### 3.6.1. Halo Photonics StreamLine

Wind vectors are calculated from raw ".hpl" VAD scan files using the ACTRIS-cloudnet halo-reader tool (Leskinen, 2023) that determines the least squares solution for the wind components from the radial velocity measurements (Päschke et al., 2015). The Manninen et al. (2016) background noise offset correction method is used by halo-reader to reduce the SNR threshold, thus increasing the amount of usable data. The correction is applied to each StreamLine (not StreamLine XR) deployment and uses the hourly background correction ".txt" raw files. The wind profile retrievals are saved as an intermediate L1 data product.

The QC steps applied to L1 data consider SNR thresholds, minimum valid range gate, wind retrieval statistical error, "despeckling" of remaining noise (Table 5). For the SNR thresholds, Manninen et al. (2016) thresholds are used to remove clearly erroneous (*flag_suspect_retrieval_removed*) and suspect (*flag_suspect_retrieval_warn*) values. The

253 thresholds are applied to the mean signal intensity within a VAD scan. VAD scan rays with SNR > 0.0055 (−22.6

254 dB) are rejected prior to averaging. This results in the L2 dataset.

During installation, an instrument bearing (from true north) needs to be entered. This can be determined by field

surveys (e.g. hard target reference, compass corrected from magnetic north) but may later be revised if a more accurate

survey is undertaken. The raw data will still have the original bearing adjustment, requiring a wind direction offset

correction. To account for this, a final manual adjustment to the instrument bearing is done at L2 for a number of the

StreamLine deployments (Table 4, bearing correction: final harmonised).

**Table 5:** StreamLine-specific quality control (QC) applied at level 2 (L2) processing stage. QC steps are carried out in row-order
(i.e. *flag_suspect_retrieval_removed* first).

| Flag name | Thresholds and steps |
|---|---|
| *flag_suspect_retrieval_removed* | • RMSE > 3 m s$^{-1}$ between observed scan points and fitted wind. Threshold based on manual inspection.<br>• Fewer than 75% of scan rays have SNR > 0.0055 (−22.6 dB).<br>• Range gates below 45 m. Threshold based on manual inspection across all instruments.<br>• Based on intercomparisons, PAROIS lower range gates are found to have unrealistic wind speed bias (Section 4.2, Appendix 3). |
| *flag_suspect_retrieval_warn* | • RMSE > 2 m s$^{-1}$ between observed scan points and sine-wave fitted wind. Threshold based on manual inspection.<br>• Despeckle: if < 3 consecutive range gates have valid wind retrievals for one timestep, all 3 range gates flagged. Threshold based on manual inspection.<br>• Based on intercomparisons, PAROIS lower range gates are found to have unrealistic wind speed bias (Section 4.2, Appendix 3). |
| *flag_low_signal_removed* | • Average SNR across all scan rays < 0.0055 (−22.6 dB). "tentative threshold" (Manninen et al., 2016). |
| *flag_low_signal_warn* | • Average SNR across all scan rays < 0.007585 (−21.2 dB). Reliable post-background correction threshold (Manninen et al., 2016). |

### 3.6.2. WindCube WLS70

QC and harmonisation of the WLS70 data here starts with the L1 product wlscerea_1a_windLz1Lb87M10mn-

HR_v02. The wind field products are derived from DBS scans internally by the manufacturer firmware. The output

is averaged to 10 min and text files are converted to standardised NetCDF using the *raw2l1* python code (Drouin,

2022). The L1 data availability is reported for each 10 min interval and a QC step is included to ensure a minimum

of 80 % of data have a sufficient signal at each range gate. The WLS70 reports a carrier-to-noise ratio (CNR) which

is the ratio between the detected signal power and the wideband noise power in the Doppler spectrum (Vaisala,

2022) used to reject retrievals with CNR < -31 dB.

Here, the L1 product undergoes further QC steps to create the L2 product (Table 6). The L1 10 min data availability

variable is used to flag suspect intervals as *flag_suspect_retrieval_warn* and *flag_suspect_retrieval_removed*. As

manual inspection shows sporadic unrealistic retrievals at altitudes above ~700 m agl, these are removed using

vertical and easterly wind thresholds (Table 6) with corresponding timesteps flagged

*flag_suspect_retrieval_removed* (Table 6).

**Table 6:** WLS70 specific quality control (QC) applied at level 2 (L2) processing stage. QC steps are carried out in row-order (i.e. flag_suspect_retrieval_removed first).

| Flag name | Thresholds and steps |
|---|---|
| *flag_suspect_retrieval_removed* | • 10 min interval data availability < 10 %.<br>• Erroneous high-altitude retrievals: Vertical wind < 2.5 m s$^{-1}$ & easterly wind component < 1 m s$^{-1}$ and range > 750 m. Thresholds based on manual inspection. |
| *flag_suspect_retrieval_warn* | 10 min interval data availability > 10 % and < 75 %. |
| *flag_low_signal_removed* | No QC applied at L2. Applied internally in L1 wlscerea_1a product only. |
| *flag_low_signal_warn* | No QC applied at L2. Applied internally in L1 wlscerea_1a product only. |

### 3.6.3. Vaisala WindCube Scan 400s

QC and harmonisation of the L1 data product uses 2s temporal resolution w400s_1a_LqualairLzamIdbs_v01 data (Céspedes et al., 2024). Its wind profiles are based on a rolling calculation through the dataset's time dimension, updated after each DBS line of sight scan.

The L1 data are used to create a L2 dataset at 1 min temporal resolution. The first round of valid DBS scans in the L1 data are found by sub-setting the data by an existing internal L1 flag *wind_speed_status*. Further suspect or erroneous retrievals are filtered using a moving window approach along the time dimensions (Appendix 1) which assigns *flag_suspect_retrieval* flags. As with the WLS70, the low signal thresholds are already applied internally by the manufacturer firmware and then within the w400S L1 product where CNR below −20 dB and above 5 dB are excluded (Céspedes et al., 2024).

### 3.7. Level 3 (L3) data harmonisation across instruments

The L2 data from each instrument (Sect. 3.5) are brought together as the final harmonised dataset provided in Network Common Data Form (NetCDF) file format and processed as follows:

- To have a common vertical dimension that is consistent horizontally, the vertical dimension is adjusted to height above sea level (NetCDF dimension name "altitude") which is obtained from the known range gate, station elevation and scan angles.

- To have a common vertical resolution, the eastward and northward wind components (*u, v*) are resampled to 25 m height by linear interpolation (Steinheuer et al., 2022). The maximum interpolation is between two range gates of the individual sensor (Table 4, range gate length). If data are unavailable causing this distance to be exceeded, the wind components are set to a missing value. Where resampled heights contain multiple L2 QC flags (Table 7) the maximum flag value is assigned.

- To have consistent vertical extent of data availability between sensors, the maximum altitude is 6500 m, defined by the w400S valid retrieval extent.

- To have a common time dimension, the range-resampled data are analysed at regular intervals. Two harmonised time intervals are available (600 s and 3600 s). The time labels assigned indicate the end of the time integration period in UTC e.g., for the 600 s interval, 03:00 UTC is derived from data between 02:50:01 and 03:00:00 UTC.

• The percentage occurrence of each L2 QC flags is determined for each time interval (Table 7).

• Mean u ($\bar{u}$) and v ($\bar{v}$) wind components are calculated at each time interval, from which the horizontal wind speed

($W_s$) and direction ($W_d$) are calculated:

$$W_s = \sqrt{\bar{u}^2 + \bar{v}^2}, \qquad (1)$$

$$W_d = \arctan\left(\frac{-\bar{u}}{-\bar{v}}\right) 180/\pi, \qquad (2)$$

with $W_d$ adjusted across 0 – 360°:

$$W_d = \begin{cases} W_d + 360, & W_d \leq 0 \\ W_d, & W_d > 0 \end{cases} \qquad (3)$$

• With data aligned along the same time and altitude dimensions, a third and final `station` dimension is then added

as a measurement location identifier.

• Deployment attributes (Table 4) are added (e.g. system_id, Table 7) to differentiate deployments at an individual

station.

• Each file contains one day of data and are named *paris_dwl_L3V{version}_ {first}_{last}_{resolution}s.nc* with

*first* and *last* timesteps (format: YYYYMMDDHHHHMM), the temporal *resolution* (s) and processing *version*

(format: e.g. 1.21).

**Table 7:** Content of the daily NetCDF files which contain the harmonised data product for all stations. Quality control flags are a
percentage occurrence of L2 QC flags (Sect. 3.5) per time interval. Data have 1, 2 or 3 dimensions (#-d). For 3-d data these are
time, height and station. For 2-d they are time and station. The NetCDF standard name and units are given as attributes for each
NetCDF variable **(Eaton et al., 2024)**.

| NetCDF *standard_name* (variable name) | #-d | Description *(see text for details)* |
|---|---|---|
| time | *1* | Timestamp: end of time interval. 600 s and 3600 s time intervals are provided, in separate data files (600 s e.g. 00:00:01 → 00:10:00 and 3600 s e.g. 00:00:01 → 01:00:00). All variables are harmonised to this resolution as averages (e.g. wind) or percentage occurrence (e.g. flags) |
| altitude | *1* | Altitude of centre of each measurement gate above mean sea level (m). Harmonised gates are 25 m from 0 – 6500 m with values linearly interpolated to this resolution |
| station | *1* | Measurement location identifier, all are listed even if no valid data are retrieved during the file's date. |
| eastward_wind (u) | *3* | Mean eastward wind component (m s⁻¹) using all valid samples within time interval |
| northward_wind (v) | *3* | Mean northward wind component (m s⁻¹) using all valid samples within time interval |
| wind_speed (ws) | *3* | Horizontal wind speed calculated from eastward_wind and northward_wind (m s⁻¹) (Eqn 1) |
| wind_from_direction (wd) | *3* | Horizontal wind direction calculated from eastward_wind and northward_wind (degrees from true north) (Eqn 2) |
| system_id | *2* | Serial number of sensors deployed at station at a given time |
| latitude (station_lat) | *1* | Latitude of the measurement station (degrees, decimal, WGS84) |
| longitude (station_lon) | *1* | Longitude of the measurement station (degrees, decimal, WGS84) |
| station_altitude | *1* | Average height of station above sea level (reference_geoid: EGM96) (m) |
| station_height | *1* | Measurement station height above ground level (m). Ground level is the "street" level so if the station is on a rooftop, the height will account for the building height and any mounting structure |
| n_rays_in_scan | *2* | Number of rays in a scan. e.g. 12 for a VAD scan that has 12 samples within one scan |
| n_pulses | *2* | Number of pulses in a given ray. More pulses, the higher the integration time |
| raw_gate_length | *2* | Gate length prior to L3 aggregation (m) |

| flag_suspect_retrieval_warn | 3 | Percentage of values within time interval with retrieval warning not linked to low signal (*flag_low_signal_warn_pc*) or out of range (*flag_ws_out_of_range_removed_p*c). Retrievals retained but treat with caution |
|---|---|---|
| flag_suspect_retrieval_removed | 3 | Percentage of values within time interval with retrieval error not linked to low signal (*flag_low_signal_warn_pc*) or out of range (*flag_ws_out_of_range_removed_p*c). Data removed |
| flag_low_signal_warn | 3 | Percentage of values within time interval with a low signal. Retrievals retained but treat with caution |
| flag_low_signal_removed | 3 | Percentage of values within time interval with a low signal. Retrieval rejected |
| flag_ws_out_of_range_removed | 3 | Percentage of values within time interval with wind speed outside reasonable retrievable range ($> 60$ m s$^{-1}$) (i.e. removed). Evaluated after all other retrieval QC |

## 4. Data evaluation

The harmonised data are evaluated using independent *in-situ* radiosonde (Sect. 4.1) and the Eiffel Tower (Sect. 4.2) data to cover both the vertical and temporal data characteristics.

### 4.1. Radiosonde vertical profiles

To evaluate the vertical component of the wind retrievals, Windsond S1H2-R radiosondes (Sparv Embedded AB, Linköping, Sweden) were released. They consist of a Styrofoam enclosure tethered to a helium balloon (circumference 123 cm, 5 m thread length). The lightweight radiosounding systems (22.9 g, including sensor, battery and balloon) can be released from within urban areas (subject to air traffic control approval) and are able to measure wind speeds between $0 - 150$ m s$^{-1}$ and wind direction ($0 - 360°$) every 1 s as they ascend through the atmosphere (Sparv Embedded, 2019). The wind speed and direction are derived from the GPS position of the sonde every 1 s with a resolution of 0.1 m s$^{-1}$ and 0.1°. The measurement accuracy is *ca*. 5 % for wind speed, whilst the wind direction accuracy depends on the GPS conditions (Sparv Embedded, 2019). The sondes transmitted to a Sparv RR2 radio receiver and the data is logged to a Windows laptop with Sparv WS-250 software.

Six radiosondes were released at Parc André Citroën (PABPAC, 48.84165 °N, 2.27416 °E) on Nov 22 2023, from 16:45 – 17:57 UTC and on Nov 23 2023, from 06:47 – 10:11 UTC. The first day had predominantly clear skies, whilst the second day was overcast with intermittent light rain. Both days had low ground-level wind speeds that increased to up to 10 m s$^{-1}$ until 1 km asl (Fig. 3) and winds ranging from northerly to westerly wind directions (Figure 4). Observed ascent speeds of $1.7 \pm 0.4$ m s$^{-1}$ until 1 km asl translated to flight durations of approx. 10 min. Horizontally, the radiosondes travelled between 2.0 and 4.7 km during their flight time (Figure 1). For the comparison statistics (Table 8), the DWL and sonde data were matched based on the time of closest horizontal distance between the respective DWL and sonde location.

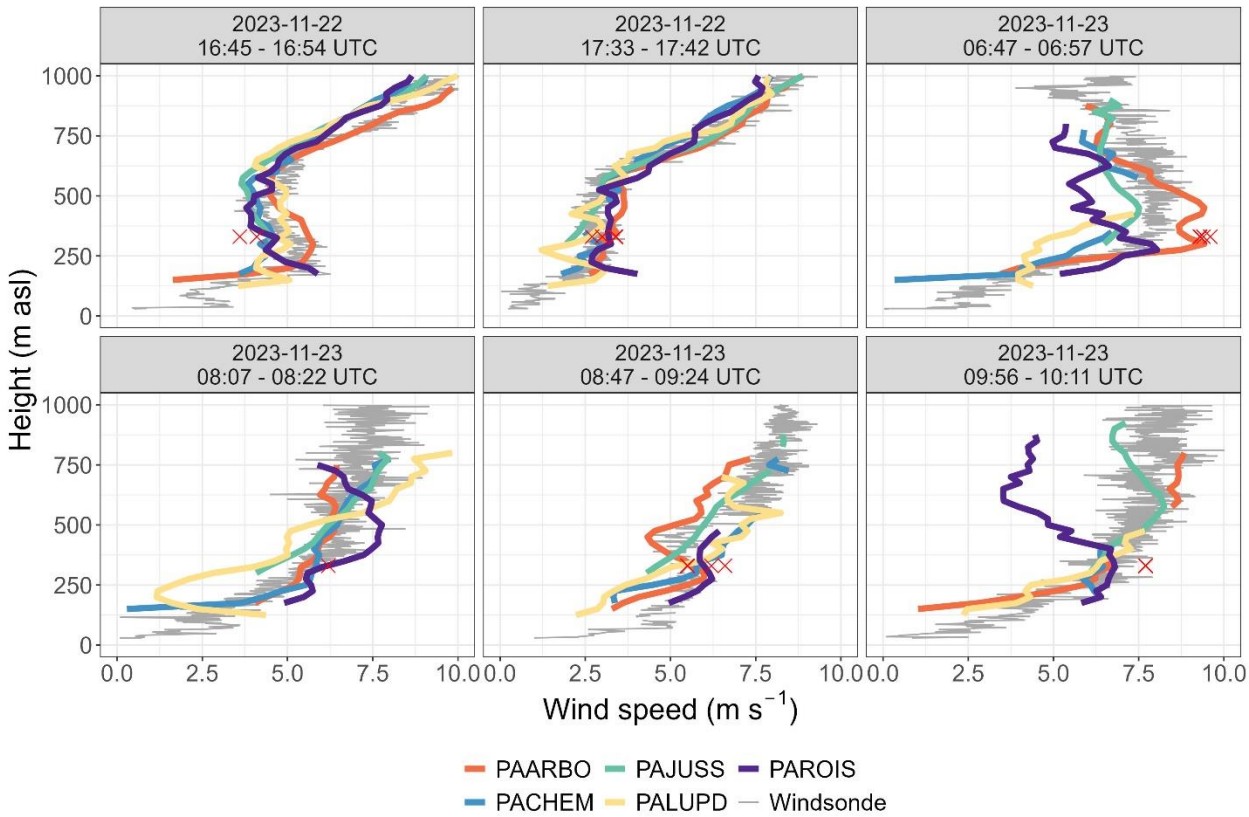

**Figure 3**: Horizontal wind speed observed with L3 DWL (colour, key), 2D sonic anemometer (Ultrasonique Thies compact) Eiffel Tower (red crosses, Sect. 4.2, Table 9) measurements and Windsond S1H2-R radiosondes (grey) at six times (November 22, 2023 from 16:45 – 17:57 UTC and on November 23, 2023 from 06:47 – 10:11 UTC) up to 1000 m above sea level (asl). Comparison statistics in Table 8.

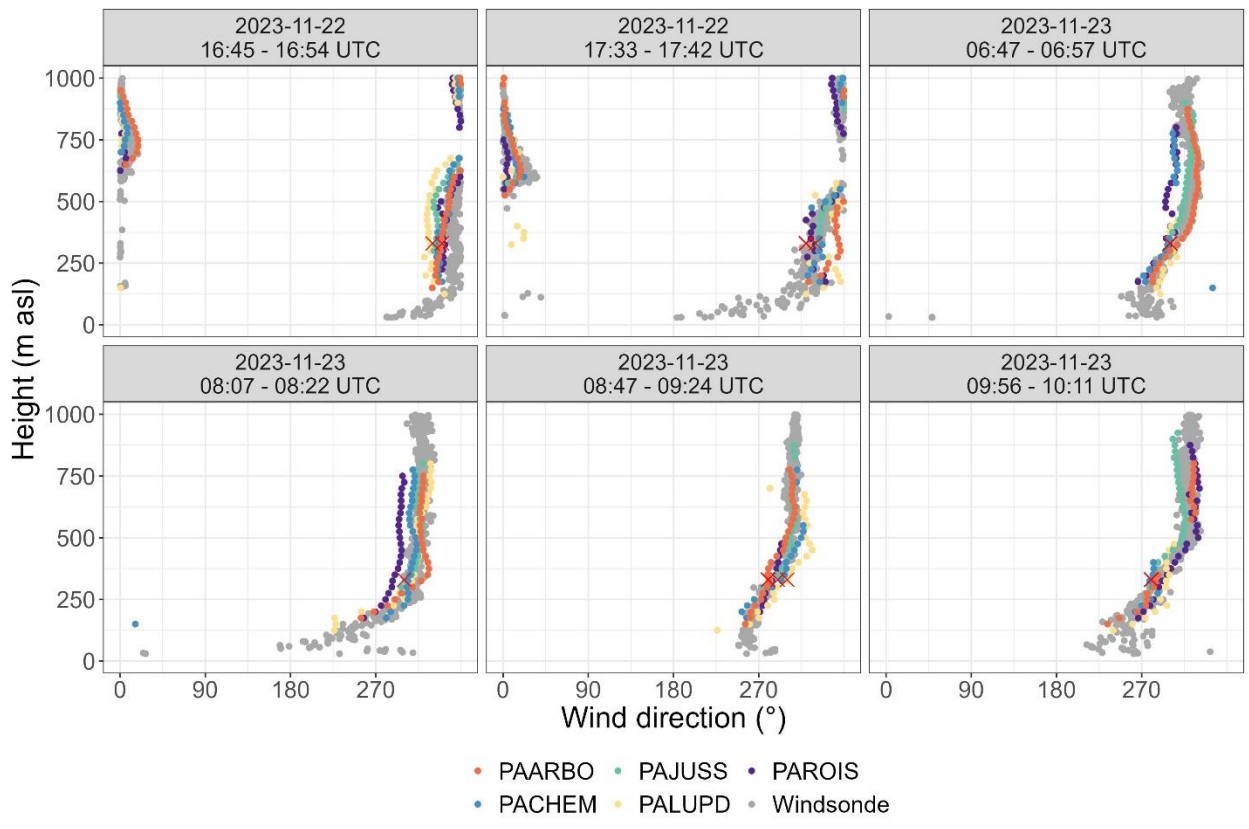

**Figure 4**: As Figure 3, but horizontal wind direction.

**Table 8. Comparison between launched radiosonde sensors and doppler wind lidars for all (*n*) matched profiles (as**
**visualised in Figure 3 and Figure 4) analysed using the mean bias error (MBE, units of variable), root mean square error**
**(RMSE, units of variable) and Pearson correlation coefficient (PCC, dimensionless).**

|  |  | Wind direction (°) | | | Wind speed (m s⁻¹) | | |
|---|---|---|---|---|---|---|---|
|  | *n* | MBE | PCC | RMSE | bias | PCC | RMSE |
| PAARBO | 165 | -0.94 | 0.95 | 10.03 | -0.05 | 0.87 | 0.92 |
| PACHEM | 134 | 2.65 | 0.89 | 17.07 | -0.37 | 0.86 | 0.96 |
| PAJUSS | 149 | 4.63 | 0.95 | 9.31 | -0.36 | 0.91 | 0.79 |
| PALUPD | 150 | 0.15 | 0.87 | 17.72 | -0.42 | 0.80 | 1.30 |
| PAROIS | 158 | 7.15 | 0.94 | 12.89 | -0.51 | 0.58 | 1.56 |

## 4.2. Eiffel Tower and Parc Montsouris in-situ time series

The two long-term Météo-France stations Eiffel Tower and Parc Montsouris have in-situ Ultrasonique Thies compact
2D ultrasonic anemometers providing 6 min mean data (Table 9, Appendix 2). The Eiffel Tower sensor is located at
321.5 m above ground level. The instrument has no surrounding obstacles and the data are not filtered for wind
direction. The DWLs are between 4.6 and 24.7 km from the Eiffel Tower (Table 7) but as the height of comparison
for all observations is well above the influence of local roughness elements, we assume all are capturing the similar
general flow, and therefore the Eiffel Tower is informative for evaluating the DWL retrievals.

During IOP1 period (Figure 2), on the 11[th] August 2023 inter- and intra-station differences in profiles of wind speed (Figure 5) and direction (Figure 6) are evident. The wind profiles are generally consistent with the Météo-France in-situ data, except for the PAROIS DWL data below around 250 m asl, where much higher wind speeds are observed. The maximum DWL retrieval height varies through the day as aerosol loading changes within and above the ABL.

Comparison of the harmonised DWL and Eiffel Tower wind speed measurements for July 2023 – March 2024 are generally consistent (Figure 7), PAROIS has the largest mean bias error (MBE 1.1 m s$^{-1}$). On closer inspection there is an unrealistic positive wind speed bias at lower range gates, supported by intercomparison with other profilers (Appendix 3). All PAROIS DWL wind speed and direction retrievals below 210 m agl (322 m asl) are therefore removed with *flag_suspect_retrieval_removed*, and flagged as *flag_suspect_retrieval_warn* for heights between 210 m and 270 m agl.

Wind direction is compared but the mean absolute error (MAE) is calculated only for periods when Eiffel Tower wind speeds > 2 m s$^{-1}$ as wind direction uncertainty increases rapidly with low wind speeds (Manninen et al., 2016; Newsom et al., 2017). The mean absolute error in wind direction is below 2° for each DWL dataset (Figure 8). The highest data frequency (reds, Figure 8) in the expected south-westerly wind direction is confirmed for all instruments.

**Table 9:** Attributes of the Météo-France station in-situ horizontal wind speed and direction evaluation data (https://www.aeris-data.fr/catalogue). The data creators are Météo-France (https://meteofrance.fr) and AERIS (https://www.aeris-data.fr). Dataset source details available (Appendix 2).

| | Eiffel Tower | Parc Montsouris |
|---|---|---|
| Dataset name | 75107005_TOUR-EIFFEL_MTO_6MIN_2023.nc | 75114001_PARIS-MONTSOURIS_MTO_6MIN_2023.nc |
| Dataset product version | 1.00 | 1.00 |
| Sensor type | Ultrasonique Thies compact | Ultrasonique Thies compact |
| Height of sensor above sea level (m) | 330 | 102.5 |
| Height of sensor above ground level (m) | 321.5 | 25.5 |
| Latitude (°N), Longitude (°E) | 48.8583, 2.2945 | 48.821311, 2.336733 |
| Closest DWL (distance, bearing) | PAJUSS: 4.6 km, 105° | PALUPD: 3.3 km, 80° |
| Farthest DWL (distance, bearing) | PAROIS: 24.7 km, 45° | PAROIS: 26.0 km, 30° |
| Temporal resolution (average, sample rate unknown) (Météo-France, 2023) | 6 min | 6 min |

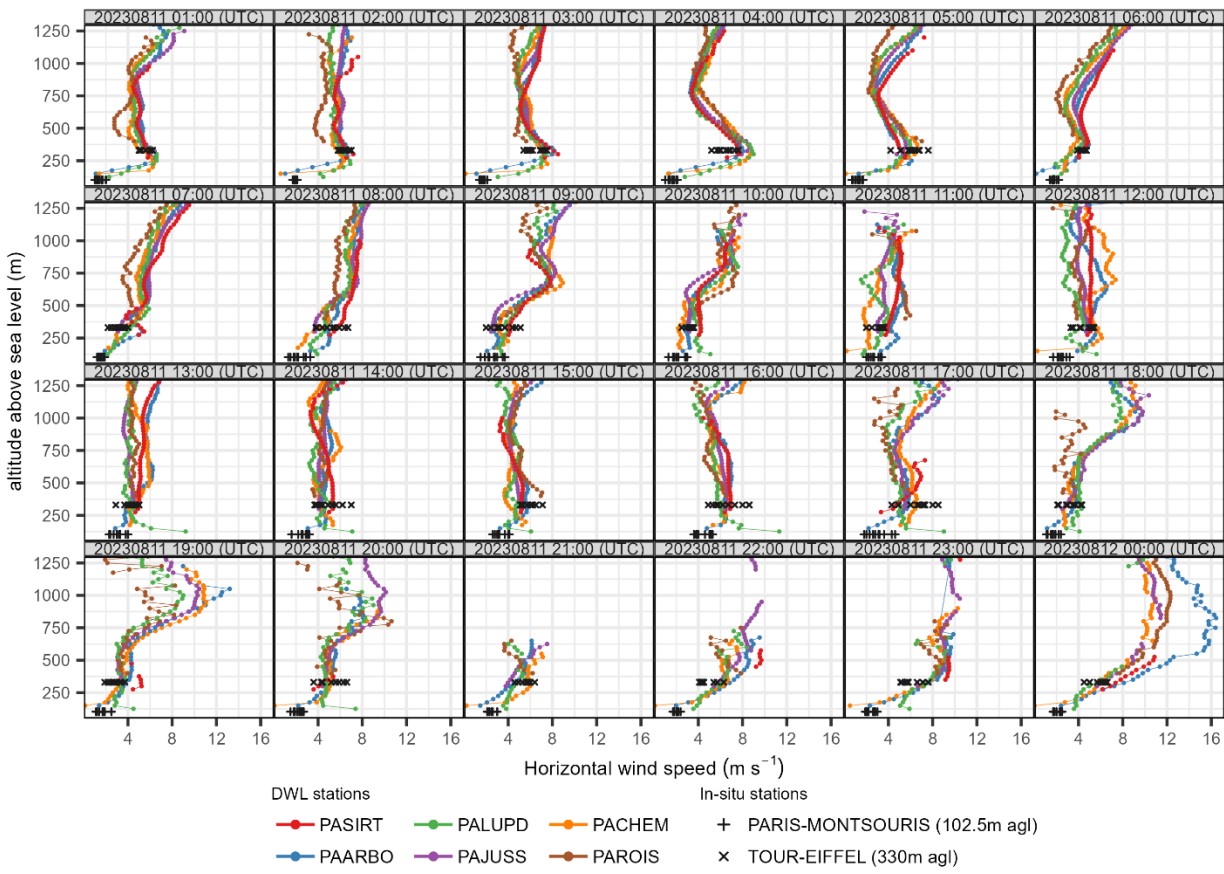

**Figure 5:** Hourly level 3 (L3, harmonised) mean wind speed observed above six DWL stations for August 11, 2023 and two in-situ stations (Table 9).

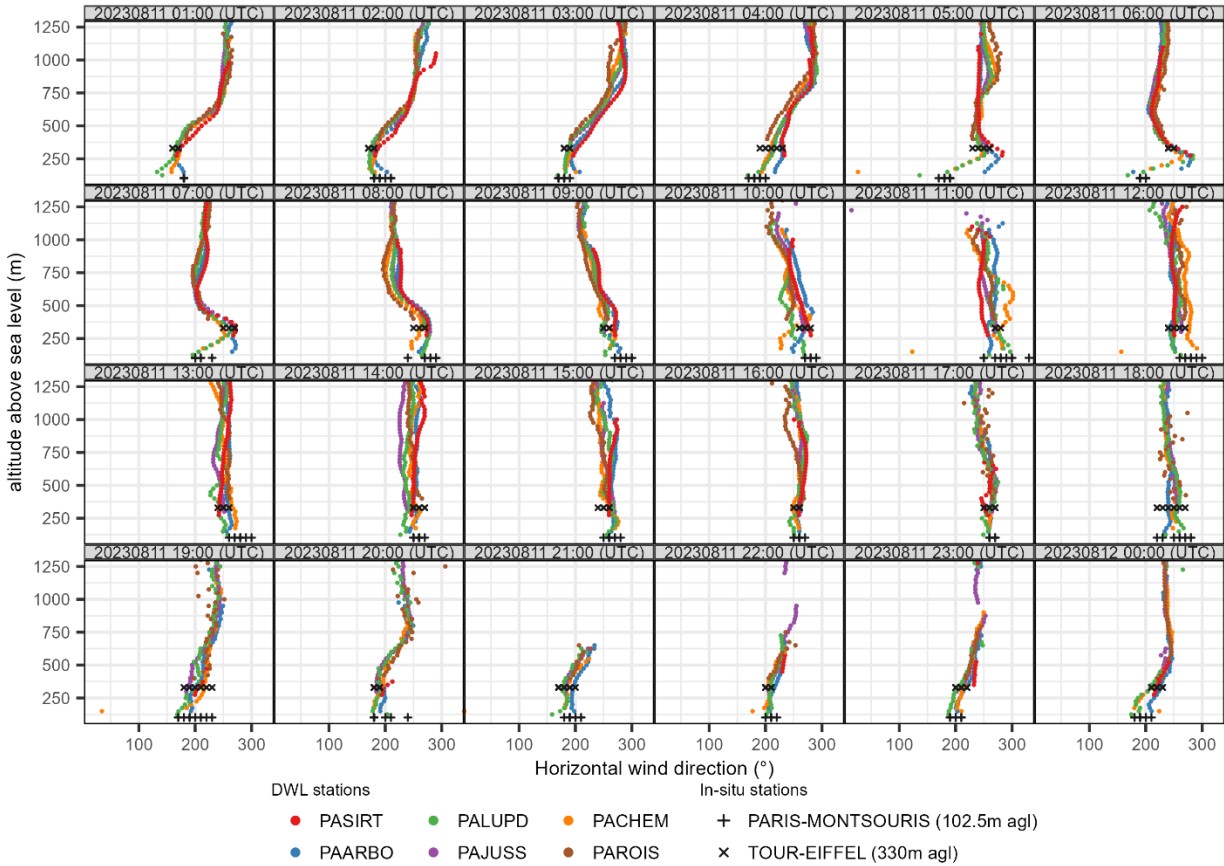

**Figure** 6: As Figure 5 but wind direction.

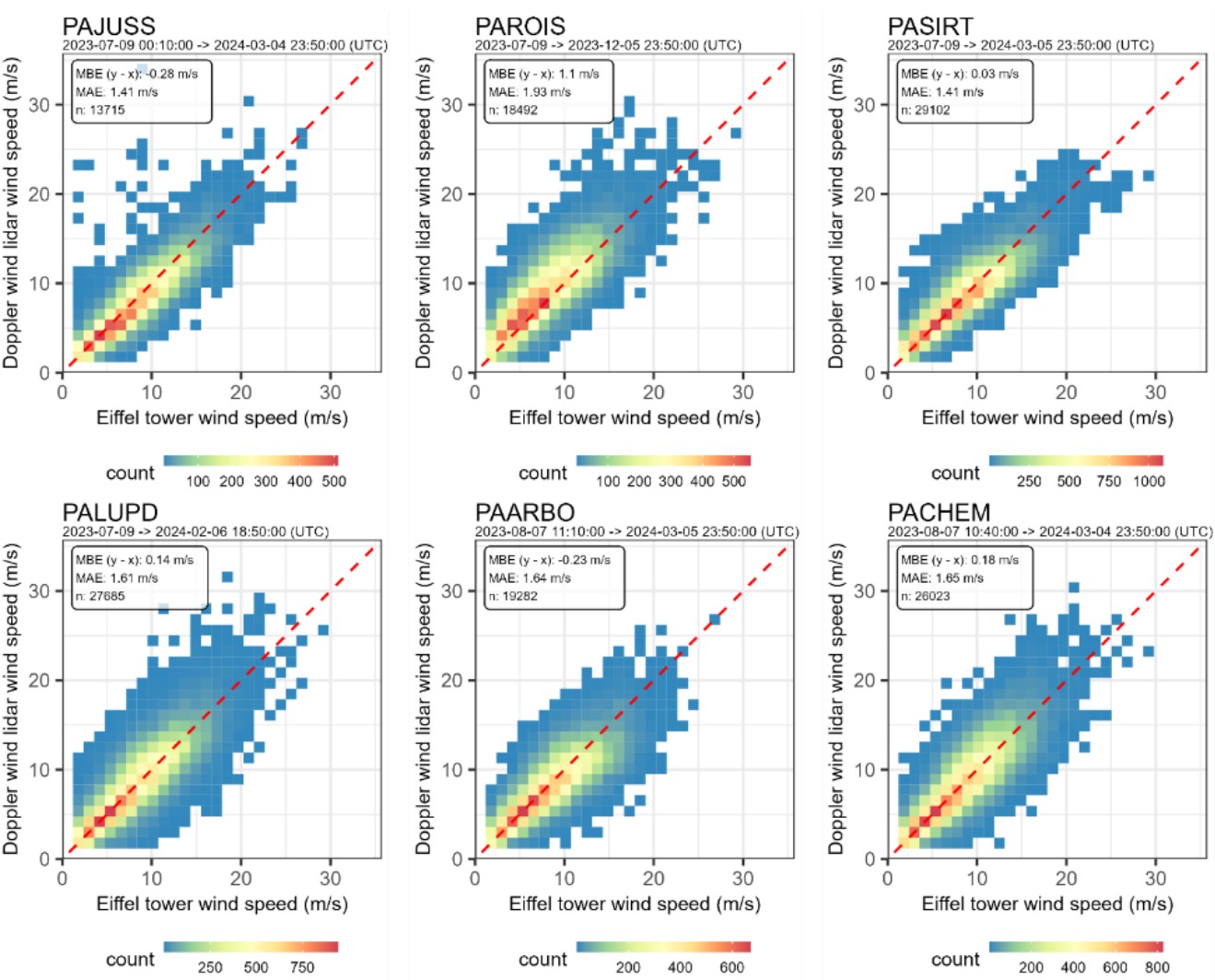

**Figure 7**: Comparison of Eiffel Tower (330 m asl, 360 s time interval, Table 9) resampled to 600 s by nearest neighbour and
harmonised (a-f) Doppler Wind Lidar (325 m asl, 600 s time interval) wind speed for July 2023 – March 2024 with mean bias error
(MBE), mean absolute error (MAE), number of period (n), density of data (colour bar, note differs between subplots) and 1:1 line
(red dashed). The data availability differs between DWL stations (subtitles, Figure 2).

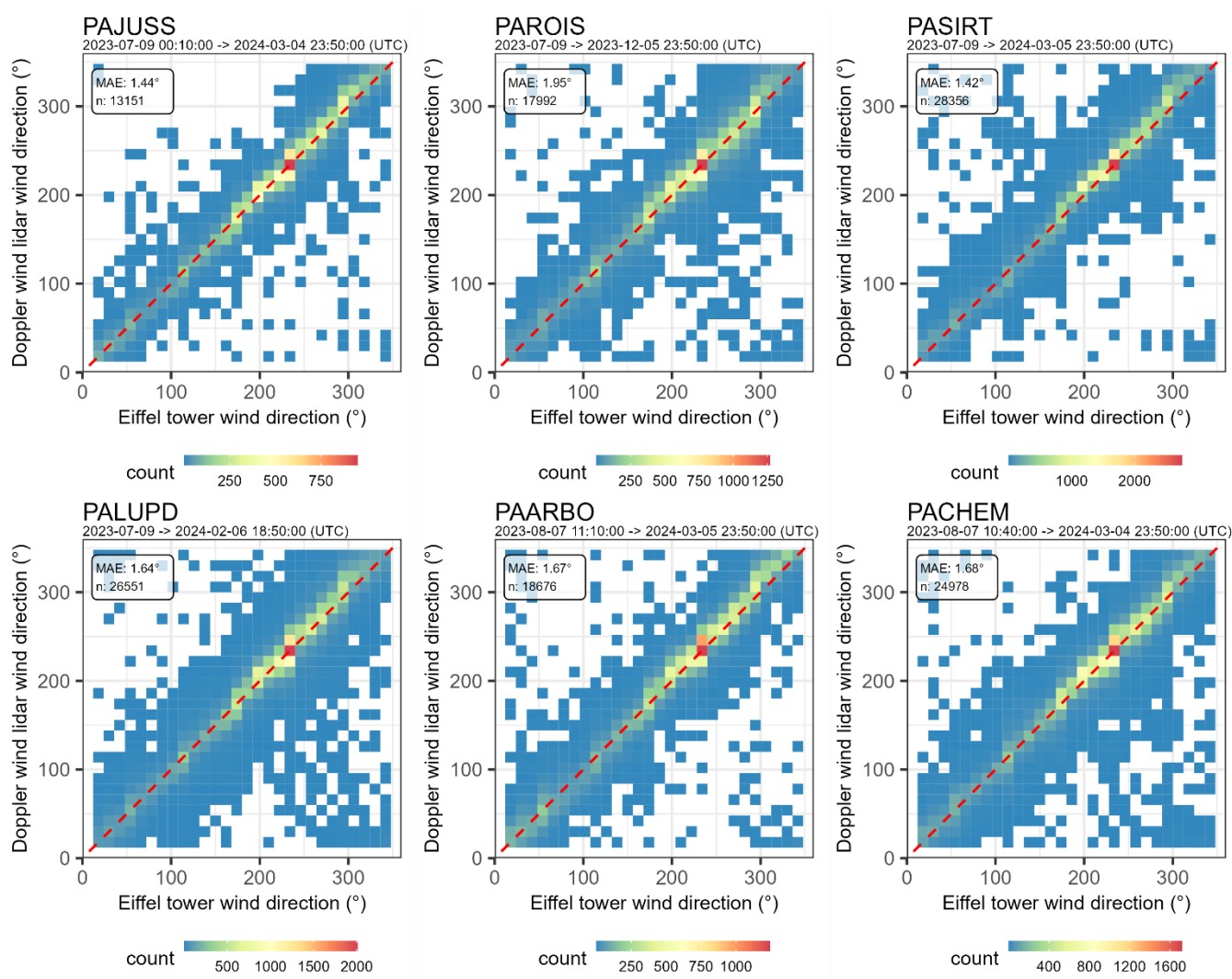

**Figure 8**: As Figure 7, but for wind direction and MAE only calculated when the Eiffel Tower wind speed > 2 m s$^{-1}$.

## 5. General guidance for data users

To support use and interpretation of the harmonised data, we provide the following guidance, addressing strengths, known limitations, and recommended practices:

- Data availability is broadly split into the EOP and IOP operational periods (Section 3.3), however within these periods the dataset is not complete through altitude, time, and instrument due to quality control filtering and atmospheric changes (signal strength, clouds). Care should be taken with e.g. sampling bias for stations with more complete datasets.

- The lowest asl retrieval depends on the instrument model and deployment asl altitude. The city centre site (PAJUSS) has the highest minimum retrieval altitude of all instruments, with the first valid wind profiles available from 275 m asl. This can be evaluated by the nearby deployment at PALUPD (first retrieval ~125 m asl) during the intensive observation period (IOP). The PASIRT site is on a plateau and is the highest asl

| 409 | | deployment altitude (154 m asl). This significant topographical feature is expected to influence the wind |
| 410 | | field. |

- For single-station studies, PAJUSS and PASIRT may be more robust given their longer time series. However,
- trade-offs include PAJUSS not sampling close to the ground and PASIRT having both the lowest maximum
- range and range resolution.
- Transect or gradient studies benefit from using IOP data when all stations were operational. The trade-offs
- can be evaluated with respect to the instrument and instrument deployment specifications (Table 3 and Table
- 4, respectively) and data availability analysis (Figure 2).
- The harmonised dataset extends to 6500 m asl to accommodate retrievals from PAJUSS. Generally, there is
- limited data availability (< 5 %) from other instruments between ~4000 and ~6500 m asl. Data availability
- is broadly governed by lidar signal returns, which are reduced during precipitation, thick cloud, or under very
- low aerosol conditions. Retrievals generally stop at cloud base.
- Data are not collected with a common sample rate but have been harmonised to the same time resolution.
- PASIRT and PAJUSS have multiple samples aggregated within a 10-minute interval; all StreamLine
- instruments have one sample within a 10-minute interval with PAROIS operating a unique scan timing with
- an assumed ±5-minute uncertainty (Table 4).
- The harmonisation of the height grid to 25 m by linear interpolation is a standard approach and the impact
- on a complex wind profile (a low-level jet event) is evaluated (Figure **11**). This analysis shows that the height
- grid harmonisation performs well. Level 2 data are available on request for very fine-scale urban boundary-
- layer process studies, further evaluation of the height grid interpolation, etc.
- A unified system of quality control (QC) flags is included in the dataset. Users are strongly advised to consult
- these QC flags, which indicate potential retrieval issues due to signal strength, instrument errors, and other
- factors. The flagging system is deliberately transparent and enables users to exclude suspect or low-signal
- retrievals for stricter analyses.
- Each station intentionally samples a different urban, suburban or rural settings (Table 1), and their
- representativeness will be influenced by local surface roughness, orography, and direction of the approaching
- flow.
- Wind speed and direction retrievals below 322 m asl at PAROIS (Roissy Airport) have been removed due to
- a technical issue resulting in positive wind speed bias at lower range gates.

## 6. Data availability

The harmonised L3 data described here are available at https://doi.org/10.5281/zenodo.14761503 (Morrison et al.,
2025). Table 7 gives the attributes of the daily NetCDF files. Météo-France observations are available from thredds-
su.ipsl.fr AERIS catalogue (https://thredds-su.ipsl.fr/thredds/catalog/aeris_thredds/catalog.html) with access details
in Appendix 2.

## 7. Code availability

The code used to retrieve the wind from the StreamLine instruments is available at the GitHub repository www.github.com/actris-cloudnet/halo-reader (details Sect. 3.5) that on 16 Feb 2024 merged to "doppy" https://github.com/actris-cloudnet/doppy. This code was adapted for production of this dataset. The adapted fork of the code is available here https://github.com/Urban-Meteorology-Reading/halo-reader. The code used for the remaining data production is available here https://github.com/Urban-Meteorology-Reading/paris-harmonised-dwl. The data visualisation code is available on request.

## 8. Conclusions

Boundary layer wind profile data from six doppler wind lidar (DWL) stations deployed along a 40 km transect through Paris, France are harmonised for the period 06/2022 – 03/2024. The dataset consists of a long-term extended observation period (EOP) and two intensive observation periods (IOP1 and IOP2) with different data availability. The EOP has fewer operational sensors but longer temporal coverage suited for long-term urban-rural study. The IOP has 5 months when all six DWL stations are operated, making it suited for studies of intra-urban effects.

Here we provide a harmonised dataset, which has removed inter-instrument heterogeneity by creating a common set of both three-dimensional (time, height, station) properties and quality control (QC) flags for data status (reject, suspect, use). The harmonised data comprehensive evaluation includes temporal analysis with the Eiffel Tower mounted sonic anemometer data. There is excellent agreement with all DWL data. The largest biases are for the DWL deployed at Roissy Airport (station PAROIS, mean bias error 1.1 m s$^{-1}$), likely attributable to the near field lower surface roughness. Vertical consistency is evaluated with a radiosonde campaign during IOP2. These indicate good overall consistency with height. The implementation of the retrieval and quality control steps has allowed independently validated wind profiles to be combined in one ready-to-use dataset, which is designed to expedite the use of DWL observations in a broad range of urban climate studies and model evaluation.

## Appendix 1. WindCube Scan 400S L2 suspect retrieval removal QC

The L1 w400s_1a_LqualairLzamIdbs_v01 dataset includes multiple scan types within the time series (e.g. not DBS scans) and some erroneous/unrealistic scans not removed during L1 quality-control (QC) steps. As the L1 *wind_speed_status* flag designed to select the realistic DBS scans did not identify all unrealistic retrievals, here a further QC step is applied with aim of including only realistic DBS scans in the L2 dataset.

To remove the unrealistic DBS and the non-DBS retrievals, for each range gate in each 30 s interval the median wind speed is calculated. If the wind speed is > 60 m s$^{-1}$ for more than 1 % of all range gates within the 30 s interval, all 2 s values within that 30 s interval are rejected.

**Appendix 2. Météo-France data source and access methods**

Météo-France in-situ wind observations were found by searching for the relevant station via the https://www.aeris-data.fr/catalogue/ interface, in the subsection "METEO-FRANCE, 6 minutes data from ground-based stations (RADOME and extended network)". The dataset IDs are DatasetScanAERISTHREDDS/actrisfr_data/cbe74172-66e4-4e18-b2cc-31ad11ed934d/2023/75107005_TOUR-EIFFEL_MTO_6MIN_2023.nc (Eiffel Tower) and DatasetScanAERISTHREDDS/actrisfr_data/cbe74172-66e4-4e18-b2cc-31ad11ed934d/2023/75114001_PARIS-MONTSOURIS_MTO_6MIN_2023.nc (Parc Montsouris). The data access URL is https://www.aeris-data.fr/catalogue/?uuid=cbe74172-66e4-4e18-b2cc-31ad11ed934d.

**Appendix 3. Evaluation of PAROIS wind speed bias**

**Evaluation with Doppler SODAR at PAROIS**

Horizontal wind speed retrievals from doppler wind lidar (DWL) 30 at PAROIS are suspiciously high at lower range gates. To evaluate this, November 2022 and August 2023 data from a nearby Doppler SODAR (model PCS.2000-64/MF, METEK GmbH) with 10 m vertical resolution are used. The SODAR is located next to the northern runway of Paris Charles de Gaulle Airport 115 m agl, within 2 km of station PAROIS. The SODAR data are available every 10 minutes on regular, rounded schedule (e.g. 01:00, 01:10).

For each DWL (level 2) height level, the SODAR data with the closest matching height is identified. The wind speed data from both instruments are aligned in time using nearest neighbour approach. Analysis is restricted to periods when both datasets have valid, quality-controlled measurements available. The SODAR data are filtered for low signal to noise ratios.

For each height, the wind speed difference (SODAR minus DWL) is computed for all coincident 10-minute averages throughout the August period. The mean bias error (MBE) and standard deviation are calculated to evaluate the bias of the DWL with respect to the SODAR (Figure **9**). For example, the August 2023 MBE range by height is from $-5.0 \pm 3.2$ m s$^{-1}$ at the lowest evaluated height (55 m asl, 3945 samples) to $-0.6 \pm 2.2$ m s$^{-1}$ at the highest (220 m, 328 samples, 761 samples at the second highest). Similar MBE are seen in November 2022, suggesting a long-term issue.

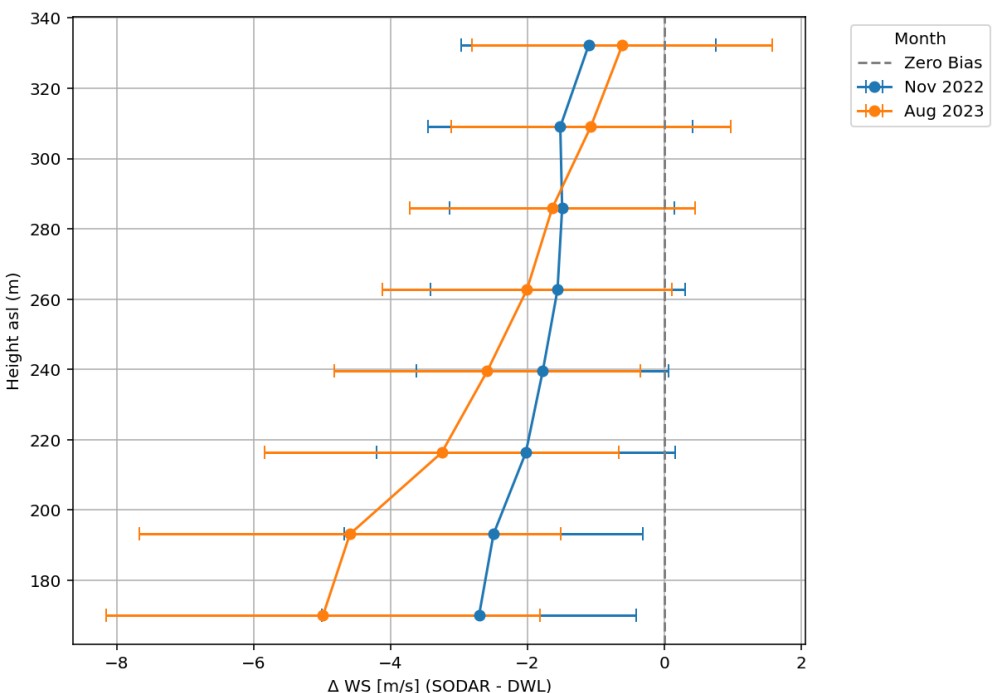

Figure 9. Wind speed as a function of height above sea level (asl) mean bias error (points, SODAR minus doppler wind lidar serial
number 30 at PAROIS), with standard deviation (whiskers) of differences for two months (colour).

**Evaluation with another StreamLine in Bristol, UK, 2025**

To evaluate the DWL (StreamLine 30) bias for range gates above the SODAR retrieval range (> ~ 200 m agl, Figure
9), an independent StreamLine  (DWL#03) not used in Paris (serial number 03, operated by University of Reading)
are compared during an ongoing (2025) NERC ASSURE/ERC urbisphere measurement campaign Bristol, United
Kingdom. Here, DWL30 is installed in the city centre (BRTOBA latitude 51.442°, longitude -2.614°, 31 m asl)
approximately 1 km from DWL03 at BRHARB (51.449°, -2.624°, 11 m asl). DWL03 was serviced by the
manufacturer prior to this installation.

This second pairwise comparison, shows the same general bias (Figure 10) for the 17[th] June 2025 – 14[th] July 2025
period analysed.  The MBE decreases to < 1 m$^{-1}$ above 210 m agl, and < 0.5 m s$^{-1}$ above 270 m agl.

As a result of these analyses, DWL30 L1 data are rejected for heights agl < 210 m with
*flag_suspect_retrieval_removed*; and flagged as *flag_suspect_retrieval_warn* for heights agl between 210 m and 270
519 m. See Table 5 for StreamLine quality control flagging details.

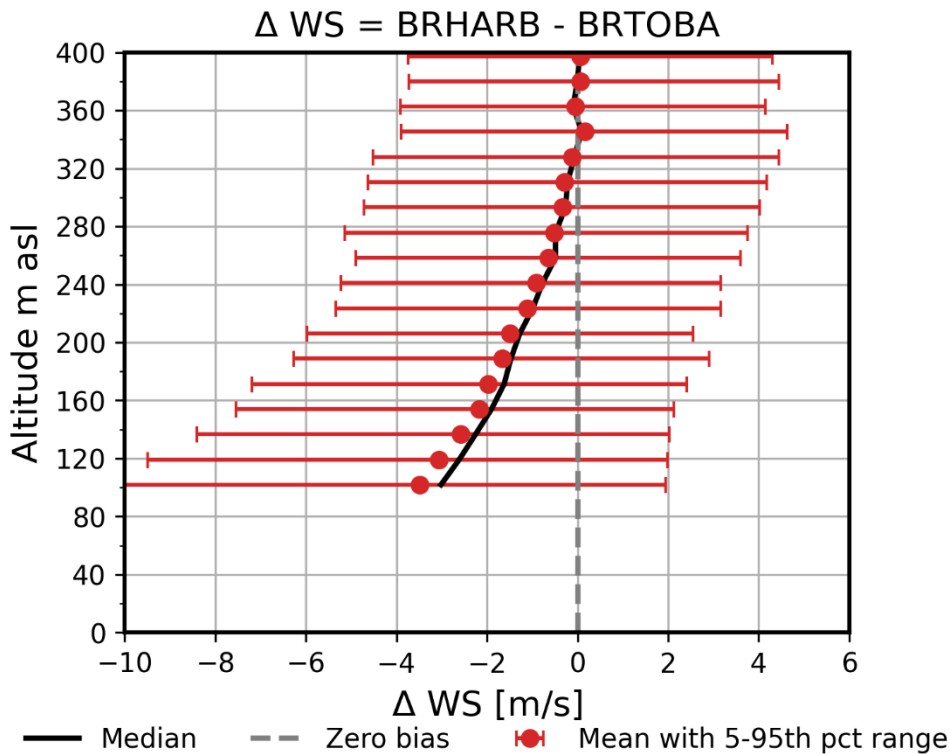

Figure 10. Mean bias error in wind speed between StreamLine doppler wind lidars installed in Bristol, United Kingdom at station BRHARB (serial number 03) and BRTOBA (serial number 30) as a function of height above sea level (asl) with (points) the mean bias at each evaluated height and (horizontal bars) the $5^{th}$ – $95^{th}$ percentile range of differences between $17^{th}$ June and $14^{th}$ July 2025.

**Appendix 4. Impact of vertical coordinate resampling to coarse vertical grid: low-level jet example**

The data harmonisation process from level 2 (L2) to level 3 (L3) involves resampling the height coordinates to a common grid. This may affect fine-scale variability, particularly near the surface or in cases with sharp vertical gradients. To demonstrate this we compare L2 and L3 data for a low-level jet (LLJ) event detected by DWL serial number 175 at PACHEM on August $24^{th}$ 2023 (Figure **11**).

Resampling linearly interpolates between L2 data points (circles - lines, Figure **11**), to obtain the L3 resampled values (crosses, Figure **11**), shown up to ~600 m asl. The LLJ core wind speeds are above 10 m s$^{-1}$ at 298 – 333 m asl between 02:00 – 06:00, giving more than one L3 vertical grid point. The LLJ core height – determined as the height of the maximum retrieved wind speed – has an absolute difference due to vertical resampling of up to 15.33 m at 04:00, with corresponding differences in wind speed (direction) of 0.07 m s$^{-1}$ (5.45°) (Table 10).

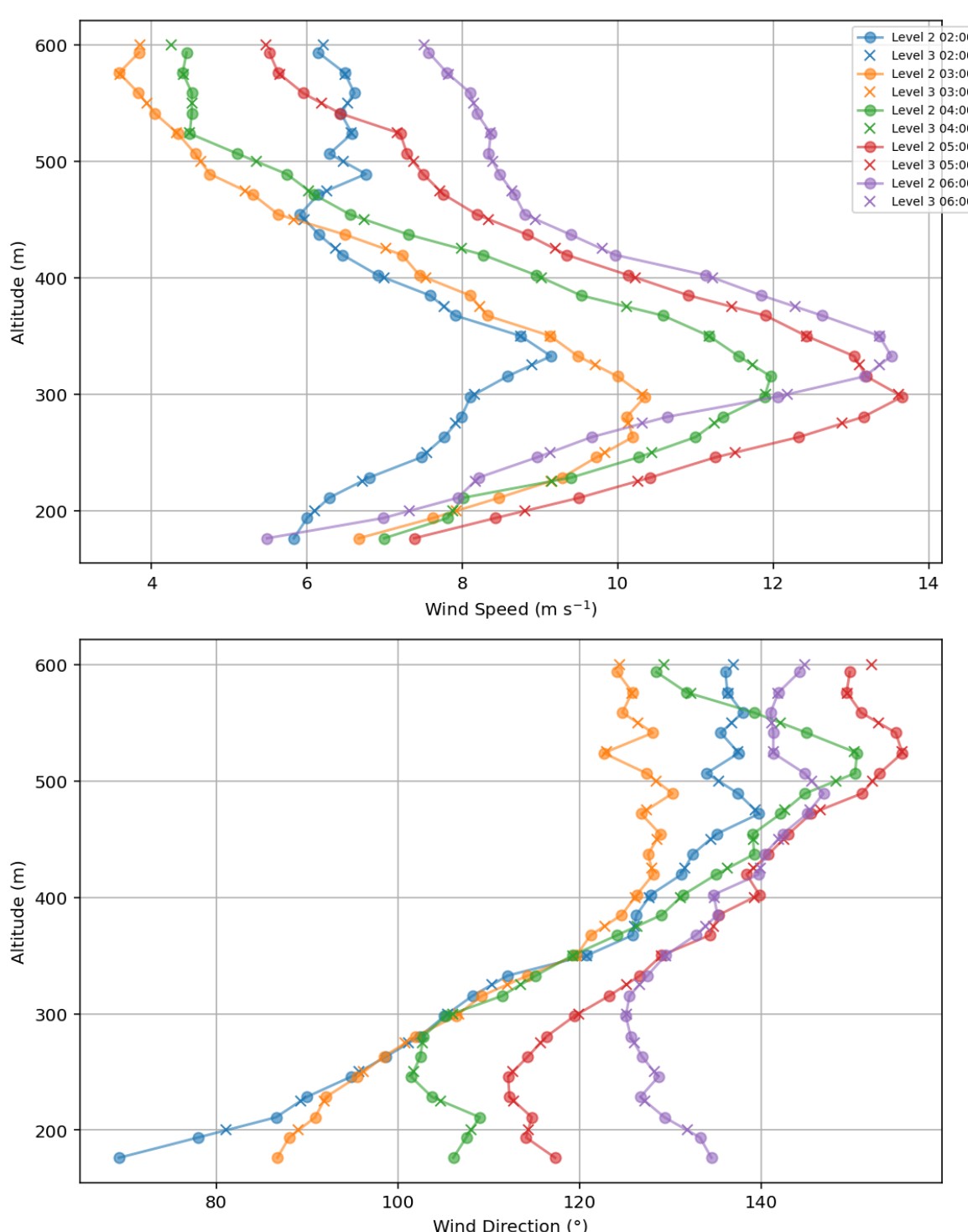

Figure 11. Comparison of (top) wind speed and (bottom) wind direction profiles retrieved from PACHEM doppler wind lidar serial
number 175 for level 2 data at original heights (circles) connected by straight lines (shown), and level 3 data (crosses at resampled
heights) for six full scans (no temporal aggregation) on morning of August 24[th] 2023. See Table 10 for statistics.

**Table 10. Statistics (for Figure 11 case) for maximum wind speed derived from level 2 and level 3 data, and associated altitude and wind direction.**

| Time (UTC, 24 Aug. 2023) | Max wind speed (m s$^{-1}$, L2) | Altitude of max wind speed (m, L2) | Wind direction (°, L2) | Max wind speed (m s$^{-1}$, L3) | Altitude of max wind speed (m, L3) | Wind direction (°, L3) | Wind speed difference (L2-L3) | Altitude difference (L2-L3) | Wind direction difference (L2-L3) |
|---|---|---|---|---|---|---|---|---|---|
| 02:00 | 9.14 | 332.72 | 112.08 | 8.89 | 325 | 110.36 | 0.25 | 7.72 | 1.72 |
| 03:00 | 10.35 | 297.95 | 106.38 | 10.31 | 300 | 106.71 | 0.04 | -2.05 | -0.34 |
| 04:00 | 11.97 | 315.33 | 111.46 | 11.9 | 300 | 106.01 | 0.07 | 15.33 | 5.45 |
| 05:00 | 13.66 | 297.95 | 119.41 | 13.61 | 300 | 119.86 | 0.05 | -2.05 | -0.45 |
| 06:00 | 13.52 | 332.72 | 127.46 | 13.36 | 325 | 126.58 | 0.15 | 7.72 | 0.88 |

**Author contribution**

Conceptualisation: WM, SG, AC. Data collection: WM, DL, JC, BC, MAD, JCD, AF, MH, VM, JM, JP, MZ. Data analysis: WM, DL. Other data processing: WM, DL, JC, MAD, AF. Harmonised data product generation and writing-original draft: WM. Writing – review & editing: All. Funding acquisition: WM, SG, AC, MG, SK. Figures: WM, DL, AC, JC. Research facilities: MH (SIRTA), VM (Meteo France).

**Competing interests**

The authors declare that they have no conflict of interest.

**Acknowledgements**

Funding for the processing and harmonization was provided by the European Commission for the "Carbon Atmospheric Tracer Research to Improve Numerics and Evaluation (CATRINE)" project (grant agreement 101135000); DWL operation and radiosounding campaigns were funded by ERC "urbisphere - coupling dynamic cities and climate" (grant agreement 855005), by ATMO-ACCESS (grant agreement 101008004, ATMO-TNA-3—0000000125) and by the European Commission through the project "Pilot Application in Urban Landscapes - Towards integrated city observatories for greenhouse gases" (grant agreement 101037319) and OBS4CLIM, Region île de France (convention no. 20007179), and the European Union's Horizon 2020 research and innovation programme project RI-URBANS (grant agreement No. 101036245). Radiosounding equipment were loaned by National Centre for Atmospheric Science (NCAS) Atmospheric Measurement & Observation Facility (AMOF) and the operation and analysis of the Bristol DWL is part of NERC NE/W002965/1 *ASSURE: Across-Scale processeS in URban Environments*. The authors thank EDF R&D/CEREA for providing data collected by the DWL WLS70. The authors thank the following staff and researchers at the University of Freiburg (Manuel Carrera, Daniel Fenner, Benjamin Gebert, Rainer Hilland, Joshua Lartey, Dirk Redepenning), the University of Reading (Matthew Clements) and at SIRTA/LMD (Florian Lapouge and Jérémie Trules) for fieldwork and data support, respectively. Météo-France RADOME observational data 75107005_TOUR-EIFFEL_MTO_6MIN_2023.nc and 75114001_PARIS-MONTSOURIS_MTO_6MIN_2023.nc are made available by the AERIS data centre and atmospheric service. Météo-France RADOME observational data 75107005_TOUR-EIFFEL_MTO_6MIN_2023.nc and 75114001_PARIS-MONTSOURIS_MTO_6MIN_2023.nc is maintained by the French national center for Atmospheric data and services AERIS.

The authors would also like to thank SIRTA and EDF R&D/CEREA for providing the lidar data used in this study; the QUALAIR-SU team (Cristelle Cailteau-Fischbach) and infrastructure for their scientific support: Qualair is an observation platform of the OSU Ecceterra of Sorbonne University operated by LATMOS with the support of IPSL; Météo-France for providing access to PAROIS (Dominique Legain); the French national centre for Atmospheric data and services ACTRIS-FR and Météo-France for providing the standard meteorology data (and metadata – Dominique Legain) used for the evaluations; Vincent Michoud and Gilles Foret (LISA/IPSL) for PALUPD site access and maintenance assistance; Dr Hugo Ricketts (AMOF) for Windsond support; staff at Arboretum de la Vallée-aux-Loups for site access (PAARBO); staff at METEK GmbH for providing Doppler SODAR data; and the ASSURE/urbisphere team involved in Bristol,

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
