# Peer review of "Harmonised boundary layer wind profile dataset from six ground- 2 based doppler wind lidars in a transect across Paris, France"

_Earth System Science Data, 2025_

## Author Comment (AC1)

**Key**

Reviewer original comment

Response to reviewer comment: final comment. Table and figure numbers refer to those in the updated manuscript

**Anonymous Referee #1**

The presented topic of the manuscript is very interesting and relevant for atmospheric boundary layer research in urban environments. The final harmonized and open data set of boundary layer wind profiles is quite valuable for the research community and opens new pathway for process based studies and evaluation of atmospheric models for urban applications.

**Specific comments:**

Line 62: "The harmonisation process involves application of wind retrievals from ..." Do you refer to application of certain wind retrieval algorithms?

As we discuss retrieval and measurement in Section 2, including previous application of wind retrieval algorithms, we clarify the sentence to be specific about what is done in this paper:

The harmonisation process involves application of a wind retrieval algorithm from to raw instrument signals data files.

Line 88: "... scanning configurations with the following parameters: azimuth ( $\theta$ ) and zenith ( $\varphi$ ) emission angles of the laser" is good but table 3 presents horizontal wind scan type with elevation angle e.g. VAD 75°, please add information that refers to this angle

Table 3 scan type column now reads

"Horizontal wind scan type and zenith (φ) angle: (# of rays per scan)"

Table values are adjusted from elevation (i.e. 75°) to zenith (i.e. 15°)

Line 233: "Päschke et al.'s (2015) retrieval method": Did you applied default values of configuration file or did you changed certain parameter settings e.g. for CNS\_PERCENTAGE? Please provide config files if possible.

The ACTRIS-cloudnet halo-reader tool is used, not any Päschke toolbox that may have a configuration file with CNS\_PERCENTAGE parameter. We modify the sentence to clarify what processing toolbox is used, and what underlying methods are used (with relevant citations).

"Wind vectors are calculated from raw ".hpl" VAD scan files using the ACTRIS-cloudnet halo-reader tool (Leskinen, 2023) that uses Päschke et al.'s (2015) retrieval method that

determines the least squares solution for the wind components from the radial velocity measurements (Päschke et al., 2015)."

Line 322: GPS position is updated every 1s?

Added "every 1 s": "The wind speed and direction are derived from the GPS position of the sonde every 1 s".

Figure 1: Please provide the time period of the presented wind rose measured at the Tour Eiffel

This information is given next to the wind rose: 06/2022 – 03/2024. We now increased the size of the font.

Figure 2a: The figure capture states "ordered from north-east to south-west", this is true if we look from the bottom panel to the top panel. But not the other way around. May better: "ordered from north-east (lower panel) to south-west (upper panel).

Lower panel has arbitrary order, but upper panel is north-east to south-west. Clarified: "(ordered from north-east to south-west)".

Figure 2b: The horizontal lines for the very low altitudes are bit confusing. These lines show that below at a certain altitude e.g. 100 m are no data? This corresponds to the lowest reasonable gate?

Added to Figure 2 caption "The near-horizontal lines at lower altitudes indicate low/no data for the first range gates."

Table 1: City centre (CC) is given as reference, please provide latitude and longitude. The abbreviation LMD means?

"Regional location relative to PAJUSS as the city centre (CC) reference location"

LMD: changed to Laboratoire de Météorologie Dynamique in the table.

Table 2: No table entry means "not available" e.g. for the "Radial wind accuracy" or something else?

Yes. Added "unavailable" to the relevant columns in Table 2.

**Anonymous Referee #2**

The manuscript describes the development and evaluation of a harmonized multi-year, multi-instrument Doppler wind lidar (DWL) dataset from a transect of six stations across the Paris metropolitan area. The paper is generally well-structured, and the dataset is of clear value to the community. However, I have several substantive concerns and recommendations that should be addressed.

1) Section 3.6 provides some descriptions of QC procedures for each instrument model, but the rationale behind the choice of certain thresholds (e.g., SNR limits, RMSE cut-offs, range gate thresholds) is only briefly mentioned. Please elaborate on: How these thresholds were determined (empirical inspection vs. manufacturer specification vs. literature values)? Whether sensitivity tests were performed to assess the impact of varying these thresholds on data retention and accuracy?

Added "Threshold based on manual inspection" to Table 4 and Table 5 flag\_suspect\_retrieval\_warn. The basis for threshold determination for all other flags is described.

The flag\_error and \_warn system is designed to allow the user to conduct their own sensitivity analysis. In the new Section 5 "guidance to data users" we explain that this approach provides maximum transparency and flexibility for the user that may wish to apply stricter or looser QC depending on their application.

2) Although the dataset harmonizes variables and QC flags across instruments, it is not clear whether any systematic inter-comparisons or bias corrections between different DWL models were conducted before merging. Given known differences in hardware performance (e.g., range resolution, beam configuration), the authors should add more detailed description to explain: Whether cross-calibration between collocated or overlapping instruments was attempted? How residual biases between instruments might influence the spatial gradients along the transect?

Clarified in Section 3.4 "No full cross-calibration between co-located instruments was conducted due to logistical challenges in co-locating long-term EOP instruments with IOP instruments, instrument maintenance delays, and the prioritisation of maximising IOP data availability." And added to Table 3 newly sourced StreamLine wind accuracy figures from the manufacturer.

3) The transect spans sites with markedly different surface characteristics (airport, highrise urban, low-rise suburban, rural plateau). While this diversity is a strength, it also complicates interpretation. Please provide a more systematic evaluation of how site-specific surface roughness, orography, and local obstacles might influence the retrieved wind profiles. And consider including a table or figure summarizing estimated roughness lengths or urban canopy parameters for each site, with discussion of implications for representativeness.

Thank you for your suggestion. We add the land cover characteristics within a 5 km radius of each site to Section 3.2 (new Table 1) and reference the new Table 1 in the Section 3.2. site description.

We appreciate the importance of the morphometric parameters on the observed interstation differences. However, we argue that the systematic evaluation of site-specific surface roughness is outside the scope of regular ESSD articles as it would constitute further interpretation of the data. Users can now apply different models or look-up tables to estimate roughness length based on the land cover to facilitate further analysis.

4) The radiosonde comparison (Section 4.1) is limited to two release days, both during IOP2. While these examples are illustrative, they are not statistically robust. If more radiosonde launches were available during the campaign (or from nearby operational soundings), please include them in the evaluation. Provide quantitative statistics (bias, RMSE, correlation) for all matched profiles, not just visual comparisons.

Unfortunately, no additional processed radiosonde data are available beyond the two IOP2 case studies presented. The nearest operational soundings are at Trappes, located 16 km from the nearest station (SIRT) and are therefore not considered as they would certainly diverge in the lower altitudes. Given this constraint, our analysis is limited to these two releases.

We now include quantitative statistics (bias, RMSE, and correlation) for all profiles, rather than relying solely on visual comparison (Table 8 and Section 4.1).

5) Temporal evaluation with in-situ measurements: The Eiffel Tower comparison is appropriate, but: the wind speed bias at PAROIS is noted but not fully explored—please quantify whether this is a constant offset, wind-direction-dependent, or height-dependent bias.

In the new Appendix 3 the StreamLine 30 data are compared to two newly acquired independent datasets: (1) when the DWL#30 was at PAROIS is observed with a PCS.2000-64/MF SODAR located near northern runway (2) when DWL#30 is deployment in Bristol UK, during 2025 near a StreamLine sensor that was not used in Paris.

The analysis using these new datasets, finds a strong positive bias in retrieved wind speed in the lower range gates which we attribute to instrument issues. As summarised in Appendix 3 at PAROIS as "the August 2023 MBE range by height is from  $-5.0 \pm 3.2$  m s-1 at the lowest evaluated height (55 m asl) to  $-0.6 \pm 2.2$  m s-1 at the highest (220 m). Similar MBE are seen in November 2022, suggesting a long-term issue." (Appendix 3, Figure 9) and at Bristol as "The MBE decreases to < 1 m-1 above 210 m agl, and < 0.5 m s -1 above 270 m agl. ." (Appendix 3, Figure 10).

Therefore: "As a result of these analyses, DWL30 L1 data are rejected for heights agl < 210 m with flag\_suspect\_retrieval\_removed; and flagged as flag\_suspect\_retrieval\_warn for heights agl between 210 m and 270 m. See Table 5 for StreamLine quality control flagging details."

Given the new analysis (Appendix 3), we remove the Section 4.2 comment that "PAROIS has the largest mean bias error [because of the] the relatively lower roughness of the

airport runway and surroundings" and replace with "On closer inspection there is an unrealistic positive wind speed bias at lower range gates, supported by intercomparison with other profilers (Appendix 3). All PAROIS DWL wind speed and direction retrievals below 210 m agl (322 m asl) are therefore removed with

flag\_suspect\_retrieval\_removed, and flagged as flag\_suspect\_retrieval\_warn for heights between 210 m and 270 m agl."

Updated Table 5 flags *flag\_suspect\_retrieval\_removed* "Thresholds and steps" column with "Based on intercomparisons, PAROIS lower range gates are found to have unrealistic wind speed bias (Section 4.2, Appendix 3)"

We add in the new Section 5 "General guidance for data users":

"Wind speed and direction retrievals below 322 m asl at PAROIS (Roissy Airport) have been removed due to a technical issue resulting in positive wind speed bias at lower range gates."

The dataset has been updated to version 1.42 with this additional quality control, with changes propagated to zenodo, doi, and all associated urls.

Figure 4 has also been updated: the Eiffel tower wind direction (red crosses) are now correctly plotted.

6) The L3 product involves resampling to common vertical and temporal grids (Section 3.7). While necessary for harmonization, these interpolations may affect fine-scale variability, particularly near the surface or in cases of sharp vertical gradients (e.g., low-level jets). Please provide a short sensitivity analysis or example quantifying the difference between original and resampled profiles for selected cases. Discuss whether interpolation across missing range gates could introduce artificial smoothing or biases.

We agree interpolations may affect fine-scale variability. New analysis is given in Appendix 4 and a summary is given in the new "guidance for modelling communities" section.

We use a representative low-level jet morning with sharp vertical gradients (August 24th 2023) and the level 2 (non-harmonised) vs level 3 (harmonised) results in Appendix 4 with a new figure (Figure 11) and table of statistics (Table 10) supported by the following text:

"The data harmonisation process from level 2 (L2) to level 3 (L3) involves resampling the height coordinates to a common grid. This may affect fine-scale variability, particularly near the surface or in cases with sharp vertical gradients. To demonstrate this we compare L2 and L3 data for a low-level jet (LLJ) event detected by DWL serial number 175 at PACHEM on August 24th 2023 (Figure 11).

Resampling linearly interpolates between L2 data points (circles - lines, Figure 11), to obtain the L3 resampled values (crosses, Figure 11), shown up to  $\sim$ 600 m asl. The LLJ core wind speeds are above 10 m s-1 at 298 – 333 m asl between 02:00 – 06:00, giving more than one L3 vertical grid point. The LLJ core height – determined as the height of the maximum retrieved wind speed – has an absolute difference due to vertical resampling of up to 15.33 m at 04:00, with corresponding differences in wind speed (direction) of 0.07 m s-1 (5.45°) (Table 10)."

7) The dataset has significant potential for NWP, LES, and inverse modeling communities, but guidance for optimal use is somewhat limited. Suggest adding a subsection or table outlining recommended uses and limitations of the dataset, including: Appropriate spatial/temporal scales for which the data are reliable; Known limitations (e.g., reduced accuracy under precipitation, lower data availability in low-aerosol conditions); Differences in performance between instruments and sites.

**Added an entirely new Section 5 "General guidance for data users":**

8) Figures 5 and 6: The color scales and symbols are sometimes difficult to distinguish for readers; please consider improving accessibility.

We updated the colour scales to be more distinguishable.